# Deep Imputation for Skeleton data (DISK) for behavioral science

France Rose [1,2] ✉, Monika Michaluk[3], Timon Blindauer[1],
Bogna M. Ignatowska-Jankowska[4], Liam O'Shaughnessy [5],
Greg J. Stephens [5,6], Talmo D. Pereira [7], Marylka Y. Uusisaari [4] &
Katarzyna Bozek [1,2,8] ✉

Pose estimation methods and motion capture systems have opened doors to quantitative measurements of animal kinematics. While animal behavior experiments are expensive and complex, tracking errors sometimes make large portions of the experimental data unusable. Here our deep learning method, Deep Imputation for Skeleton data (DISK), uncovers dependencies between keypoints and their dynamics to impute missing tracking data without the help of any manual annotations. We demonstrate the utility and performance of DISK on seven animal skeletons including multi-animal setups. The imputed recordings allow us to detect more episodes of motion, such as steps, and obtain more statistically robust results when comparing these episodes between experimental conditions. In addition, by learning to impute the missing content, DISK learns meaningful representations of the data capturing, for example, underlying actions. This stand-alone imputation package, available at https://github.com/bozeklab/DISK.git/, is applicable to outputs of tracking methods (marker-based or markerless) and allows for varied types of downstream analysis.

Animal pose tracking was largely improved by recent progress in camera precision and synchronization, which perform accurate measurements of animal movement. On one hand, infrared or visible spectrum cameras can be coupled with automated pose estimation (for example, DeepLabCut[1], LEAP[2], DeepPoseKit[3], and their refinements for three-dimensional (3D) or multi-animal tracking). On the other hand, motion capture systems directly register 3D positions of markers affixed on animals at key body parts. These pose estimation algorithms and motion capture systems estimate the position of keypoints through time, providing large quantities of data capturing animal motion. These data consist of time series of keypoints combined into a skeleton—a simplified and widely used representation of the body for downstream tasks such as action recognition or behavior analysis.

However, both motion capture systems and pose estimation models fail to determine all keypoints at all time points in an experiment (Fig. 1b). In markerless techniques combined with a posteriori tracking, insufficient lighting, blurriness, animal occlusion and automatic tracking errors result in missing or inaccurately tracked data. Some pose estimation systems, in addition to estimating each keypoint position in a given video frame, output an associated confidence score thus allowing keypoint filtering[3–8]. Marker-based techniques use motion capture cameras to directly assess the marker 3D positions and are typically more precise in determining keypoint locations[4,9,10].

[1]Institute for Biomedical Informatics, Faculty of Medicine and University Hospital Cologne, Cologne, Germany. [2]Center for Molecular Medicine Cologne (CMMC), Faculty of Medicine and University Hospital Cologne, Cologne, Germany. [3]Faculty of Mathematics, Informatics and Mechanics, University of Warsaw, Warsaw, Poland. [4]Neuronal Rhythms in Movement Unit, Okinawa Institute of Science and Technology, Onna-son, Japan. [5]Department of Physics and Astronomy, VU University Amsterdam, Amsterdam, the Netherlands. [6]Biological Physics Theory Unit, OIST Graduate University, Okinawa, Japan. [7]Salk Institute for Biological Studies, La Jolla, CA, USA. [8]Cologne Excellence Cluster on Cellular Stress Responses in Aging-Associated Diseases (CECAD), University of Cologne, Cologne, Germany. ✉e-mail: france.rose@wanadoo.fr; k.bozek@uni-koeln.de

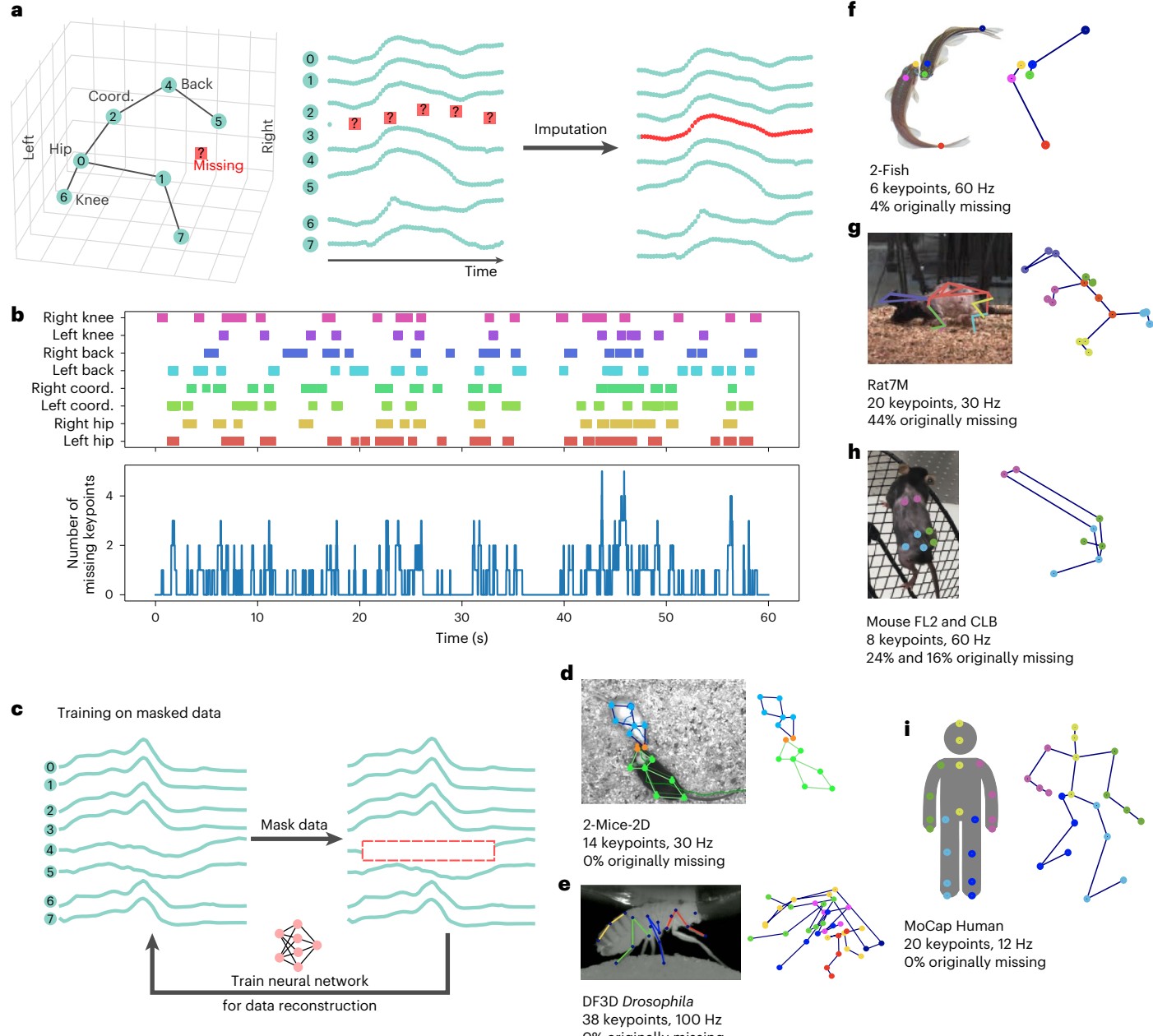

**Fig. 1 | The missing data problem. a**, DISK takes as input an incomplete sequence of a moving skeleton and imputes the missing coordinates. **b**, Example of missing data pattern from the mouse FL2 dataset. Missing segments are indicated by colored rectangles. **c**, Proposed unsupervised learning strategy: using data segments with no missing keypoints, we randomly introduce artificial gaps by masking one or more keypoints and train the neural network to reconstruct the masked coordinates. **d**–**i**, Description of the seven datasets used for the evaluation of DISK with a variety of skeletons from one to two animals: 2-Mice-2d (**d**), DF3D (**e**), 2-Fish (**f**), Rat7M (**g**), mouse FL2 and CLB (**h**) and Human (**i**). The 2-Fish (**f**), Rat7M (**g**), mouse FL2 and CLB (**h**) datasets contain original missing data in varying proportions (percentage of frames with at least one missing keypoint at the original frame rate). Coord., coordinate. Panels adapted with permission from: **d**, ref. 39, IEEE; **e**, https://github.com/NeLy-EPFL/DeepFly3D (ref. 42); **f**, ref. 43, National Academy of Sciences; **g**, ref. 4, Springer Nature America.

However marker-based techniques still fail to detect all data points due to occlusions or poor quality triangulation. Independent of the cause, the missing data result in an incomplete sequence hindering the downstream behavior analysis. Because animal behavior cannot be easily scripted and additional recordings are not always possible due to constraints in experimental design, missing data are a more pressing problem in animal compared to human behavior analysis.

Imputing missing values in general time-series data[11–15] has been developed for applications in, for example, medicine[16,17], economics[18,19]

and meteorology[20,21]. Correctly imputed values can help the final forecasting or classification task[14,15,22]. Despite these methodological developments, these methods are not commonly used in animal behavior analysis[3–8]. Skeleton data might be by nature different from other sensor data, because: (1) keypoints have physical interactions, increasing the feature correlation compared to other cases; (2) keypoints have few degrees of freedom and many coordinates' combinations are impossible due to physical constraints of the animal skeleton. On one hand, solutions for filtering of potentially inaccurate keypoint coordinates have been proposed as part of tracking packages such as Anipose,

OptiPose and Keypoint-MoSeq[8,23,24]. These solutions are, however, designed specifically to post-process markerless tracking results. These filtering methods rely on the confidence scores of the tracking methods[23] and target short-term keypoint identity switches or jitter, which represent the specific output and errors uniquely of markerless tracking methods. On the other hand, MarkerBasedImputation is a deep learning-based method proposed to post-process motion capture data[25], using recursive temporal convolutional networks (TCNs). However, it lacks a complete benchmark on multiple datasets and against other methods. To date, there is still a shortage of widely applicable methods for imputing missing keypoints in skeleton motion data, which can handle different types of animal skeletons—including one or more individuals—while remaining independent of the keypoint tracking method.

In this study, we propose Deep Imputation for Skeleton data (DISK), a method for imputing missing skeleton data in two-dimensions (2D) and 3D (Fig. 1a). In Fig. 1b, while time points with no missing data during this 1-min recording are rare, only one to two keypoints are missing most of the time, leaving the other keypoints' coordinates as a useful source of pose information. Hence, we leverage the capabilities of neural networks to use past and future information of the same keypoint, as well as the information from other non-missing keypoints, to infer missing data. We designed a fully unsupervised training scheme by introducing artificial gaps that mimic the characteristics of actual gaps in the data (Fig. 1c). These gaps are introduced according to the observed frequency of each keypoint being missing in the original data. This way DISK is trained on gaps with similar properties to those on which inference is applied.

We show the generalizability of DISK using several skeletons with varied numbers of animals, skeleton representations and behavioral tasks in 2D and 3D (Fig. 1d–h). The selected datasets (Fig. 1d–h) allow us to thoroughly test our method on different animal skeletons (mouse, fish, human, fly and rat), varying number of keypoints (from 6 to 38), original percentages of missing data (from 0% to 44%), and varying dataset sizes (Extended Data Table 1). Our method handles data from both markerless (2-Fish, 2-Mice-2D, DF3D datasets) and marker-based (mouse FL2 and CLB, Human, Rat7M datasets) systems, not relying on specific outputs from motion capture or tracking softwares but only on keypoint coordinates. Furthermore, DISK imputation is designed as a forecasting rather than a filtering procedure, enabling it to handle gaps longer than a few frames. Missing value imputation is crucial in animal experimentation, as recordings are scarce and repeated recordings of the same animal under the same setup could bias the experiment as the animal has been primed. While in markerless tracking, extensive manual annotations can allow us to obtain high coverage and tracking accuracy, the lighting conditions or angle of view can make some frames challenging or even impossible to annotate with high confidence. DISK's imputation strategy is based on learning the motion dynamics and dependencies between keypoints, which can allow us to overcome the limitations of keypoint visibility. Additionally, manual annotation is a time-consuming task, whereas DISK handles keypoint trajectory data in an automated and annotation-free manner. While controlled recordings of human actions can be more easily generated compared to animal recordings, we nevertheless demonstrate the capacity of DISK to perform on a dataset of human recordings as well. We show that imputed data indeed allow for detection of more behaviors and for improving the robustness of statistical comparisons.

Available at https://github.com/bozeklab/DISK/, DISK can be trained, evaluated and used for imputation on new datasets. In the field of animal behavior research relying on pose estimation and motion capture methods, DISK fills a gap of a posteriori imputation of missing data. Imputing keypoint trajectories in a recording will allow scientists to take full advantage of complex and costly animal behavior experiments.

## Results

### DISK uses an unsupervised learning paradigm

We chose seven datasets covering a wide range of species, behaviors and recording modalities (Fig. 1d–h). These include a 2D top-view dataset of social interactions between two mice (2-Mice-2D), a 3D markerless *Drosophila* dataset (DF3D), a 3D markerless dataset of interacting Zebrafish pairs (2-Fish), a 3D rat motion capture dataset (Rat7M), two 3D mouse motion capture datasets (FL2, open-field exploration; CLB, climbing) and the 3D CMU Human MoCap dataset (Human). Assuming that correlations between keypoint trajectories and short-term dynamics can be learned reliably by neural networks, we designed a fully unsupervised training scheme not requiring any manual annotations (Fig. 1c). Several methods are designed to perform imputation on data missing completely at random (MCAR)[13,26], that is, the value that is missing is independent of any other value, observed or not. We observed that in our datasets missing data seem to depend on the keypoint identity, the task and the camera setup. For example, in the two mouse datasets, the shoulder blade markers are missing with higher probability in the 2D exploration task (FL2 dataset) while the left hip is the most probable missing keypoint in the climbing task (CLB dataset; Extended Data Fig. 1). During training, we use segments with no missing data and introduce artificial gaps that mimic the observed missing data in the original dataset, in terms of gap length and identity of missing keypoints. The gap insertion method allows the neural network to focus during training on missing data patterns similar to the ones present at inference. Such artificial gaps are created dynamically as a form of data augmentation increasing diversity of our data independent of their original size.

### DISK correctly imputes missing skeleton data

We tested several neural network architectures—namely custom transformer (named DISK), gated recurrent unit (GRU), TCN, spatial temporal graph convolutional network (ST-GCN) and space-time-separable graph convolutional network (STS-GCN)—in the task of imputing a single keypoint on seven different skeleton datasets (Fig. 2a). As a weak baseline, we used linear interpolation as this method is widely used in behavioral science[3–8]. Root mean square error (RMSE), mean per-joint precision error (MPJPE) and percentage of correct keypoints (PCK@0.01) between the imputed coordinates and the actual coordinates within the gap were used as evaluation metrics (Fig. 2a and Extended Data Fig. 2). RMSE, expressed in normalized units and calculated on the held-out test part of each dataset, is not directly comparable between datasets because the number of keypoints, their arrangement in a skeleton and the motion range differ across animals and experiments. While MPJPE and RMSE are very close metrics, PCK provides complementary information and displays larger variations between datasets (for example, in the DF3D dataset, the tethered fly's movements are limited and most imputation errors are below threshold). Additionally, we compared the performance of DISK with Opti-Pose, Keypoint-MoSeq and marker-based imputation[8,24,25] (Fig. 2b and Extended Data Figs. 2 and 3). OptiPose builds a deep learning model to spatiotemporally refine raw 3D keypoints using different data augmentations such as masking or noise addition. Keypoint-MoSeq, whose main purpose is to identify behavior modules, contains a denoising module distinguishing between noise and real movements. Marker-based imputation uses a TCN in a recursive manner for imputation. Our method, DISK, outperforms other methods across all tested datasets (Fig. 2a,b), independent of their sizes and types of recorded animals (Extended Data Table 1). Among the tested deep learning architectures, GRU shows the second-best performance in all except the DF3D dataset, while a TCN shows the worst performance in all except the 2-Fish dataset. Surprisingly ST-GCN and STS-GCN—methods developed specifically for skeleton data—do not outperform DISK. It is important to note that linear interpolation is strongly influenced by the number of static and low movement sequences in the data (Fig. 3d,e). As a result,

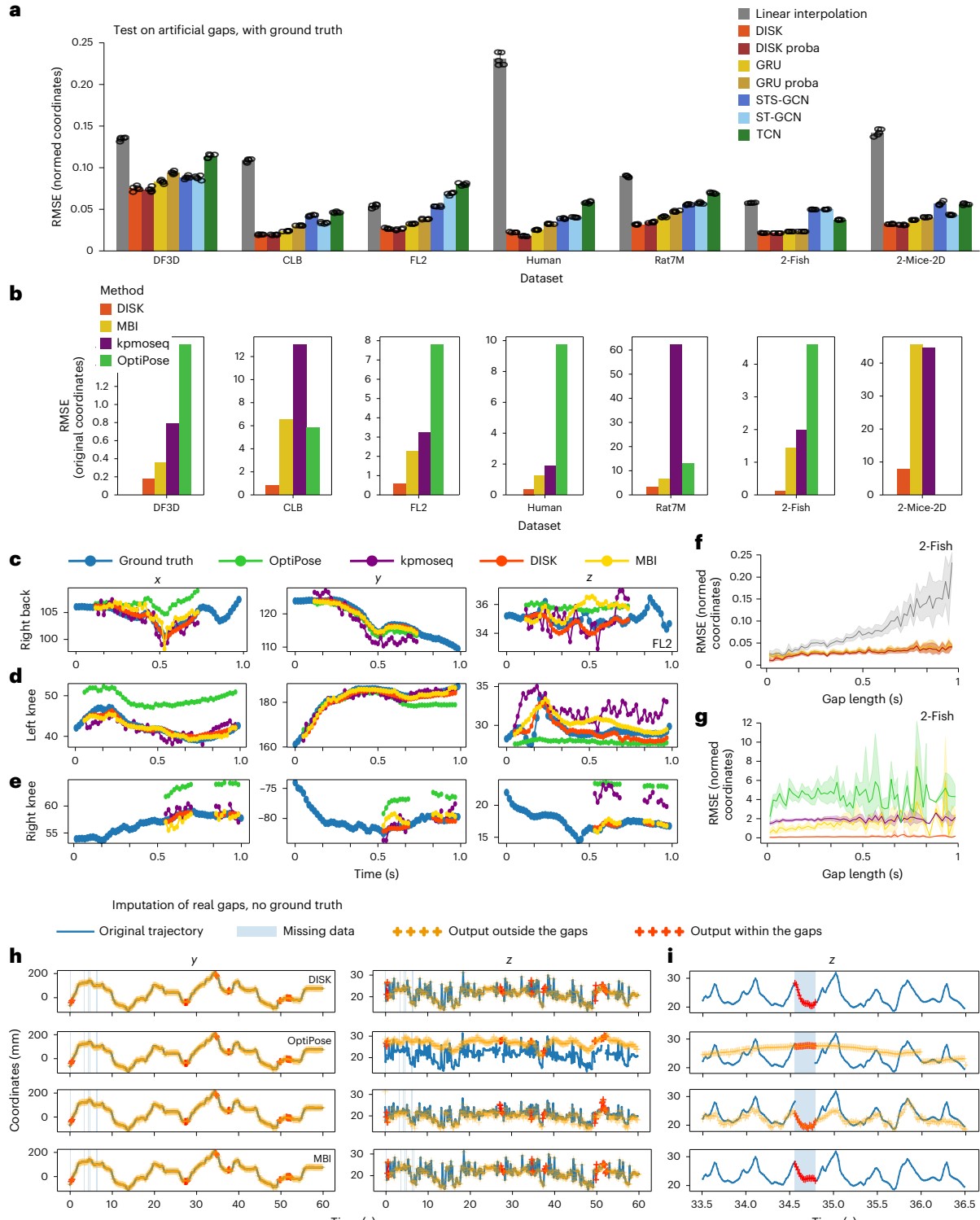

**Fig. 2 | DISK and other imputation methods' performance across datasets.**
**a**, RMSE on all datasets for all tested architectures. 'proba' refers to a modification of DISK and the GRU architecture to additionally output a confidence interval alongside the imputed values (Fig. 3). Bar plots represent the mean of five test runs of the same model, and error bars denote the s.d. Corresponding bar plots for MPJPE and PCK metrics are available in Extended Data Fig. 2. **b**, RMSE on all datasets for DISK and other tested methods, MarkerBasedImputation (MBI), Keypoint-MoSeq (kpmoseq) and OptiPose (mean RMSE of $n = 3,392$ gaps for DF3D, $n = 6,116$ for CLB, $n = 9,120$ for FL2, $n = 9,328$ for Human, $n = 26,532$ for Rat7M, $n = 13,008$ for 2-Fish, $n = 74324$ for 2-Mice-2D). OptiPose is a 3D-only method; hence no results for OptiPose are presented for the 2-Mice-2D dataset.

**c–e**, Examples of the imputation of missing coordinates of one keypoint by DISK and other tested methods (mouse FL2 dataset). Three different examples with right back (**c**), left knee (**d**) and right knee (**e**) coordinates missing. **f,g**, RMSE with respect to the gap length between tested architectures (**f**) and methods (**g**) on the 2-Fish dataset (same color scheme as **a** and **b**). The line represents the mean and the shaded area the 95% confidence interval. Corresponding plots for the other datasets are available in Extended Data Fig. 3. **h**, Output results of DISK and other methods on real gaps of one 1-min recording from the mouse FL2 dataset. **i**, Temporal zoom between 33.5 and 36.5 s of the 1-min recording in **h**. **i** is a temporal zoom of **h**. The x panels in **h** and the x and y panels in **i** have been removed to keep the figure compact.

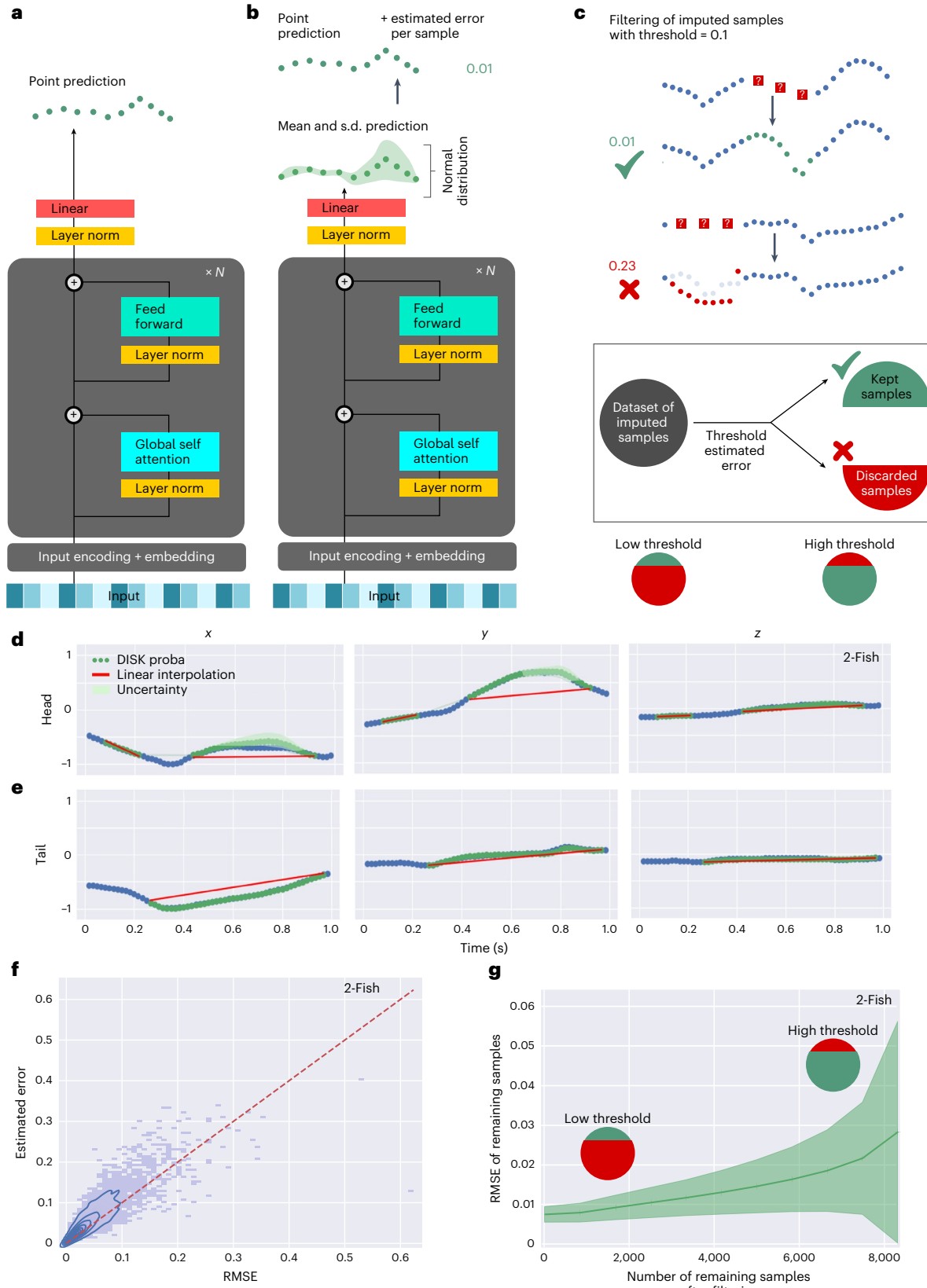

**Fig. 3 | Estimated imputation error. a**, DISK architecture for keypoint coordinate imputation. This model predicts one value for each keypoint coordinate at a given time point. **b**, DISK-proba consists of the same transformer backbone with a probabilistic head, predicting a mean value and a s.d. for each keypoint coordinate. **c**, The estimated error can be used to accept or discard samples and keep only predictions with low error for further analysis. **d**,**e**, Examples of predictions with DISK-proba on the 2-Fish dataset. Two different examples with head (**d**) and tail (**e**) coordinates missing. **f**, The estimated error per sample correlates with the RMSE per sample on the 2-Fish dataset. **g**, RMSE range of the imputed samples after thresholding the data based on their estimated error, as described in **c**. Data are presented as mean values ± s.d. *N*, number of transformer layers used in the DISK model.

linear interpolation performs well on the FL2 dataset that includes a large proportion of static sequences, whereas it performs poorly on the Human dataset that contains only actions and no resting poses. Among the methods, DISK displays the lowest error across methods and datasets by a factor of ≥2. OptiPose and Keypoint-MoSeq, designed as general post-processing and smoothing methods, capture the overall dynamics of the movements on longer timescales (Fig. 2c–e,h) but fail to impute finer movements (z-plots from Fig. 2h,i). By comparison, DISK and marker-based imputation are purely imputation methods, with a better performance across all datasets (Fig. 2b–e). While the ranking of other methods varies from dataset to dataset, DISK is consistently the best method for gap imputation.

A per-keypoint analysis (Extended Data Fig. 4) suggests that across datasets the extreme skeleton points (limb ends, head and tail) are less accurately imputed compared to core skeleton points (spine, hips, pectoral fin). This result follows the intuition that keypoints with highly correlated neighbors are easier to impute than the end nodes of the skeleton graph. In the 2-Fish dataset, the same performance is associated with each keypoint on both animals, contrary to the 2-Mice-2D dataset where this symmetry is not observed for the nose, neck and tail base. As a data-driven method, DISK performs slightly differently on each dataset, depending on the range of dynamics represented in the data, the potential difference between the left and the right side, or one and the other animal. Furthermore, our results (Supplementary Fig. 5) suggest that the imputation error tends to increase with movement speed, gap length and periodicity, and that it varies across action classes independently of their abundance in the training data. For example, errors are higher in the 'attack' class of the 2-Mice-2D dataset (2.8% of the data) and in the 'run' and 'dance' classes of the Human dataset (4.3% and 3.1%), while the least represented classes ('climb' 1.3%, and 'step' 2.7%) do not show the largest errors. DISK displays a low and relatively constant error independent of the gap length (Fig. 2f,g for the 2-Fish dataset and Extended Data Fig. 3 for the other datasets). In contrast, linear interpolation shows a major increase in RMSE and RMSE variability (95% confidence interval represented as the shaded area in Fig. 2f) with the increasing gap length, which is the reason to constrain the use of linear interpolation to small gaps[3,5]. While most tests were carried out on samples of length 60, additional tests on the 2-Fish dataset suggest that using more and longer sequences further improves performance even for shorter gaps (Extended Data Fig. 5).

Tracking errors can result in switched coordinates between keypoints and incorrect output trajectories. We implemented a 'switch data augmentation module' to increase DISK's robustness to these errors. We show that, when trained with this module, DISK maintains the same performance for samples without switched keypoints while improving the imputation in the case of switched keypoints (reduced averaged MPJPE by up to a factor of 8 in the case of the CLB dataset). The switch module improves DISK's robustness to switched keypoints independent of the identity of keypoints (within or between animals) and of the distance between switched keypoints, but with a lower corrective power as the length of the gap and/or the length of the switching increases (Extended Data Figs. 6 and 7 and Supplementary Fig. 3).

## Estimated error allows filtering of imputed data based on their quality

Black box predictions do not allow assessment of the quality of their output. As a result, it is impossible to assess the trustworthiness of the results and thus of the downstream analysis. While crucial for real-world use, there are no gold standards for uncertainty estimation of imputed data points. Existing methods[27,28] show variable confidence intervals or low correlation with true error[29]. We adapted a recent modification of a transformer architecture[30], showing good performance in estimating confidence intervals in probabilistic time-series forecasting. This modification to the transformer architecture allows the model to

output, instead of a single point prediction for each time point and each keypoint coordinate, the two parameters of a Gaussian probability distribution over the predicted values (Fig. 3a,b). The standard deviation of this probability distribution allows the visualization (Fig. 3d,e) and assessment (Fig. 3c) of the model's prediction confidence. Indeed, the standard deviation (green area in Fig. 3d,e) increases when the prediction is less accurate, reflecting the incertitude of the prediction. The average-per-sample standard deviation that we call 'estimated error' strongly correlates with the actual imputation error made by the model (Fig. 3f; Pearson correlation coefficient of 0.891 with $P < 1 \times 10^{-9}$ on the 2-Fish dataset, average of five test runs). Thus, the estimated error can serve as a quality score to filter predictions based on imputation quality (Fig. 3c,g and Supplementary Fig. 1 for the other datasets). When increasing the threshold on the estimated error, the number of selected samples increases as well as their RMSE. This probabilistic component of the model can be, in principle, added on top of any neural network backbone. However, the correlation of the estimated error with the real RMSE appears much lower for GRU, the second overall best-tested model. This correlation is the lowest in the CLB dataset for both GRU and DISK with a Pearson correlation coefficient of 0.43 and 0.746, respectively (Supplementary Fig. 1).

## DISK can impute multiple missing keypoints in single-animal and multi-animal setups

Natural skeleton poses are limited in their degrees of freedom. Observed keypoints might, therefore, carry important information for imputation of the missing ones. When increasing the number of simultaneously missing keypoints, the difficulty of the imputation task expands. We examined DISK in the task of imputing multiple missing keypoints and compared its performance to linear interpolation and GRU—our second-best-performing model. We trained one DISK model and one GRU model for each number of missing keypoints from one to seven on the mouse FL2 dataset, which contains eight keypoints in total (Fig. 4a). Both DISK and GRU show stable performance when less than half of the keypoints are missing. When more than four of the eight keypoints are missing, the performance of both models deteriorates with every additional missing keypoint. For any tested number of missing keypoints DISK outperforms GRU. In contrast, linear interpolation is agnostic to the number of missing keypoints as it is applied independently to each coordinate of each keypoint and displays a constant error range.

We further analyzed the performance of DISK in the task of imputing multiple keypoints in recordings of two animals (2-Fish and 2-Mice-2D datasets). Overall, DISK shows the best performance compared to other models in this task (Figs. 2a and 4f) and the estimated error from the DISK-proba model correlates with real imputation error (Fig. 4d,e). We compared the DISK model trained on both fish to two DISK models trained separately on fish 1 and fish 2. The model trained on the motion data of both fish showed RMSE was lower by 8% to 12% compared to the models trained on data from one fish only (Supplementary Table 2), showing that DISK is able to leverage the inter-animal dynamics to help the task of imputation (the relative improvement for the GRU model is only from 5% to 8%). However, since the keypoints are now positioned on two separate animal bodies, keypoint trajectories show a more complex pattern of correlation as the interaction between the two fish is expected to decrease with the growing inter-fish distance. When the two fish are close during chase or attack, they react in a close loop to each other[31], but when swimming at different locations in the tank, they appear not to interact. As both fish have three keypoints and are interchangeable in this analysis, we considered different combinations of missing keypoints (Fig. 4b). For short gaps (length under 30; Fig. 4c) the RMSE stays low and increases slightly with the distance between the fish independent of the number of missing keypoints and their configurations. However, for long gaps (length above 30; Fig. 4c) the RMSE increases with the distance between

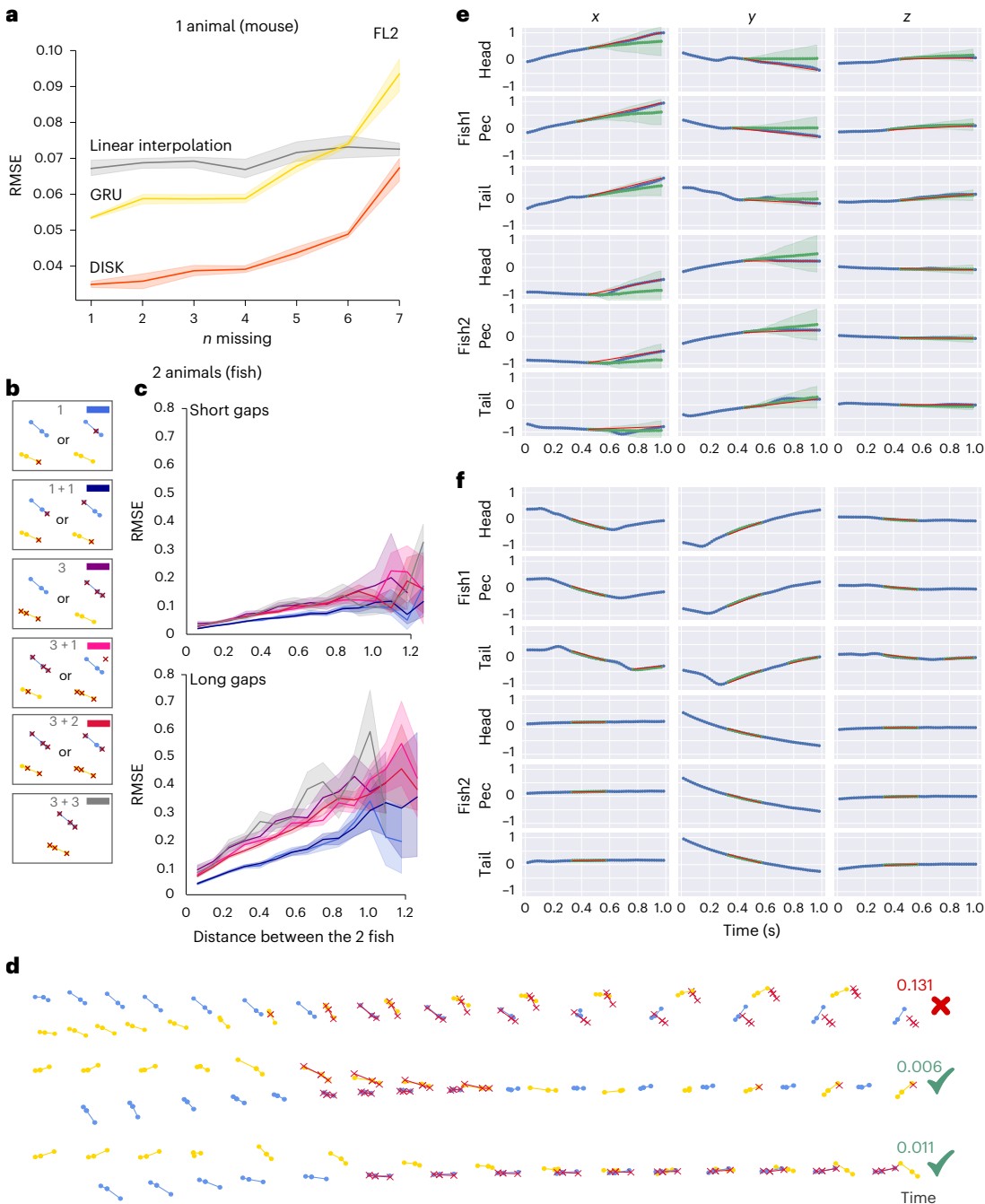

**Fig. 4 | Inference of multiple simultaneously missing keypoints. a**, Imputation performance when multiple keypoints are missing simultaneously. We trained a separate network for each count of missing keypoints. Data are presented as the mean ± s.d. of five test runs of the model. **b–f**, A DISK-proba model was trained for inference of multiple keypoints in the 2-Fish dataset with uniform probabilites. For each sample during training, a random number of keypoints was chosen as missing simultaneously; then the length of the gap was sampled from estimated observational distribution. **b**, Explanation of the legend of missing keypoints used in **c**. Both fish can have up to three missing keypoints. The two fish are interchangeable so we use the notation '$x + y$' with $x \geq y$. '3 + 1' means one fish has three missing keypoints and the other one has one missing keypoint. Color in the upper-right corner corresponds to the line colors in **c**. **c**, RMSE with respect to the distance between the two fish and the number and scheme of missing keypoints for short gaps (upper, up to 30 frames) and long gaps (lower, from 30 to 60 frames). The inter-fish distances and RMSEs are expressed in normalized units where the coordinates of each behavior sequence are normalized within the [−1, 1] range. We kept the cases 2, 2 + 1 and 2 + 2 out to avoid clutter in the display and to compare extreme cases of few or all keypoints missing (Supplementary Fig. 2). Data are presented as mean (line) and 95% confidence interval (shaded area). **d–f**, Examples of imputation of multiple simultaneously missing keypoints with estimated error. **d**, 3D trajectories (one fish in yellow and one fish in blue) and multi-keypoint imputation (red crosses). The numbers on the right correspond to the estimated error. The first sequence corresponds to **e**, and the second sequence to **f**. The last sequence shows an example with very close inter-fish distance.

the fish and most rapidly in cases with many missing keypoints (cases 3 and 3 + 3) compared to a single missing keypoint (case 1). The RMSE increase with the inter-fish distance suggests that DISK relies on interaction patterns between the two fish. While this finding might appear counter-intuitive—tracking methods generate more errors when keypoints are close to one another—DISK is not a tracking tool. Instead, DISK models the dynamics and interrelatedness of skeleton keypoints. As high correlation (and even synchronization) across individuals

occurs during free social behavior[32], such correlation facilitates the imputation task of DISK.

## DISK learns coherent symmetries and similarities between motion sequences

Prediction of masked parts of the input is a core idea behind self-supervised methods[33–38]. By learning to correct the corrupted or fill in the missing content, such methods learn meaningful representations of the data. These representations can then be used for exploratory data analysis or for supervised downstream tasks such as classification. Therefore, DISK might learn not only to impute the missing data but also to encode keypoint sequences capturing semantic information about the underlying behavior.

To test the encoding capacities of DISK, we further inspected DISK latent representations and performed action classification on the two datasets containing labels (Human and 2-Mice-2D datasets). In the 2-Mice-2D dataset, unconstrained mice are placed in a resident–intruder scheme and three behaviors are annotated (attack, investigate and mount), while remaining behaviors are labeled as other[39]. The Human dataset contains sequences labeled as wash, walk, climb or animal behavior, with the majority of sequences unlabeled. Sequences from the Human and the 2-Mice-2D mouse datasets were projected and visualized in 2D with uniform manifold approximation and projection (UMAP; Fig. 5).

The two major groups of data points correspond to a left/right symmetry of the sequences (Fig. 5b). In the Human dataset, the two groups of points show gradients in the distance between knees and shoulders, speed in both $x-y$ plane and in the $z$-direction (Fig. 5c,e). In the UMAP plots, sequences representing walking or animal behavior are clearly separated. Sequences representing washing or animal behavior are mixed, as well as those representing climbing and walking. These mixed sequences show similarities in arm and leg movements (skeleton panels in Fig. 5a). In the 2-Mice-2D dataset, the two groups split sequences by the gradients of distance between the two mice and angle of each mouse compared to the vector basis (Fig. 5f,g). Comparing the two 2D skeleton sequences from the category 'other' (top left and bottom right on Fig. 5e), we can see similar interactions between the two mice but rotated by 180 degrees. Most of the sequences categorized as 'other' show a higher distance between the two mice compared to labeled behaviors (Fig. 5e,f), suggesting their low degree of interaction.

We further trained a random forest classifier for the task of behavior class prediction based on the DISK representations. In the Human dataset, prediction of eight classes showed a balanced F1 score of 0.86 and a balanced precision score of 0.94. In the 2-Mice-2D dataset, prediction of four classes showed a balanced F1 score of 0.85 and balanced precision score of 0.87. These results indicate that indeed, through an unsupervised imputation task, DISK has learned not only fundamental differences between motion sequences like symmetries but also higher-level information characterizing different behaviors. We see similar similarity grouping in other datasets (mouse FL2 dataset; Supplementary Fig. 4).

## Imputed data enable finer analysis of drug-induced differences in locomotion

We next tested how imputation of missing data can improve downstream analysis of behavior experiments. We used a subset of the mouse FL2 dataset including recordings of ten mice performing the floor exploration task under three pharmacological conditions (PF3845 10 mg per kg body weight, referred to as drug A; MJN110 1.25 mg per kg body weight, referred to as drug B; and vehicle[40]). In the floor exploration task, step detection needs to differentiate between real locomotory steps and other leg movements. Therefore, we used kinematic rather than positional features of ankles as in ref. 9 to detect steps. A fully resolved step is defined as a velocity peak bordered with acceleration and deceleration events, reflecting takeoff and touchdown (Fig. 6a).

As in the original analysis of this dataset[9], we detected steps on both sides (Fig. 6a) and performed the step detection before and after imputation.

After imputation, we detected on average 57% more steps (paired $t$-test $P$ value = $1.7 \times 10^{-16}$ on 82 1-min recordings), bringing the number of detected steps closer to manual counts. After imputation, no recording contained fewer than 10 detected steps, and the overall distribution shifted (solid lines; Fig. 6e) toward more detected steps, with the mode increasing from 30 to 60 steps per recording. Compared to expert annotations from 22 1-min recordings (Fig. 6d), 40% of steps were still missed after imputation on average, whereas before imputation 53% were missed, with up to 79% missed in some files. The steps detected on the imputed segments are similar in shape to the original ones (Fig. 6b), underlining the quality of imputation. In the particularly challenging example of Fig. 6c with many gaps in keypoints, visibly more steps are detected after imputation. Gaps remaining after imputation are due to too many missing keypoints occurring simultaneously, a prohibitively long gap or an imputation below the chosen quality threshold. At the end of the recording, from second 45, we can see a period of immobility with no ankle movement where no steps were detected.

We next performed analysis of step features and compared it between the imputed and non-imputed data. In this analysis, we used recordings with at least 10 detected steps on each side. Without imputation, two recordings were discarded (Fig. 6e), whereas imputation allowed us to include all recordings in the analysis and to have an overall larger number of detected steps across the recordings. In Fig. 6f, we display the distribution of two features of the swing phase of detected steps. We found a significant difference (independent samples $t$-tests) in the swing stride length between vehicle and drug A after imputation ($P$ value = 0.0002705), where the comparison was nonsignificant before imputation ($P$ value = 0.05658). For the swing duration we found a higher difference between vehicle and drug B after imputation ($P$ value = 0.0004672 after imputation versus 0.01960 before). With this example analysis, we show that imputing data with DISK allows us to use a larger portion of the recorded data in the analysis and to analyze fine motions such as steps.

## Discussion

We present DISK, a general deep learning approach for imputing missing coordinates in 2D and 3D skeleton data. DISK applies to diverse datasets of single or multiple animals, covering both motion capture and markerless tracking. Spanning species from mouse, rat and zebrafish to *Drosophila* and human, the tested datasets differ not only in skeletal anatomy but also in behavioral repertoire, with human actions performed by actors and rodent behaviors recorded in unconstrained settings. A switch data augmentation module increases DISK's robustness to switched keypoints, which may arise in tracking data when two keypoints come into close proximity. Additionally, users can filter imputed sequences based on estimated error to match their required level of tracking precision.

DISK learns spatiotemporal dependencies between keypoints and their possible dynamics. In contrast to pose estimation methods[1,2], DISK leverages temporal information, the core component of behavior, for successful imputation. Adopting a self-supervised learning paradigm, DISK not only imputes missing data but also builds meaningful representations of motion sequences. These representations differentiate sequences of varying speed, orientation and limb motion, demonstrating the method's capacity to learn patterns that can be used in behavior analysis. Broadly used in computer vision and language processing, the potential of self-supervised learning in behavior analysis has not yet been fully explored. Here we lay foundations for future applications of this type of method to the study of behavior.

We tested DISK on two two-animal datasets in which the animals display a high degree of interaction. In this setup using the coordinates of both animals benefits the imputation, showing that DISK is able to

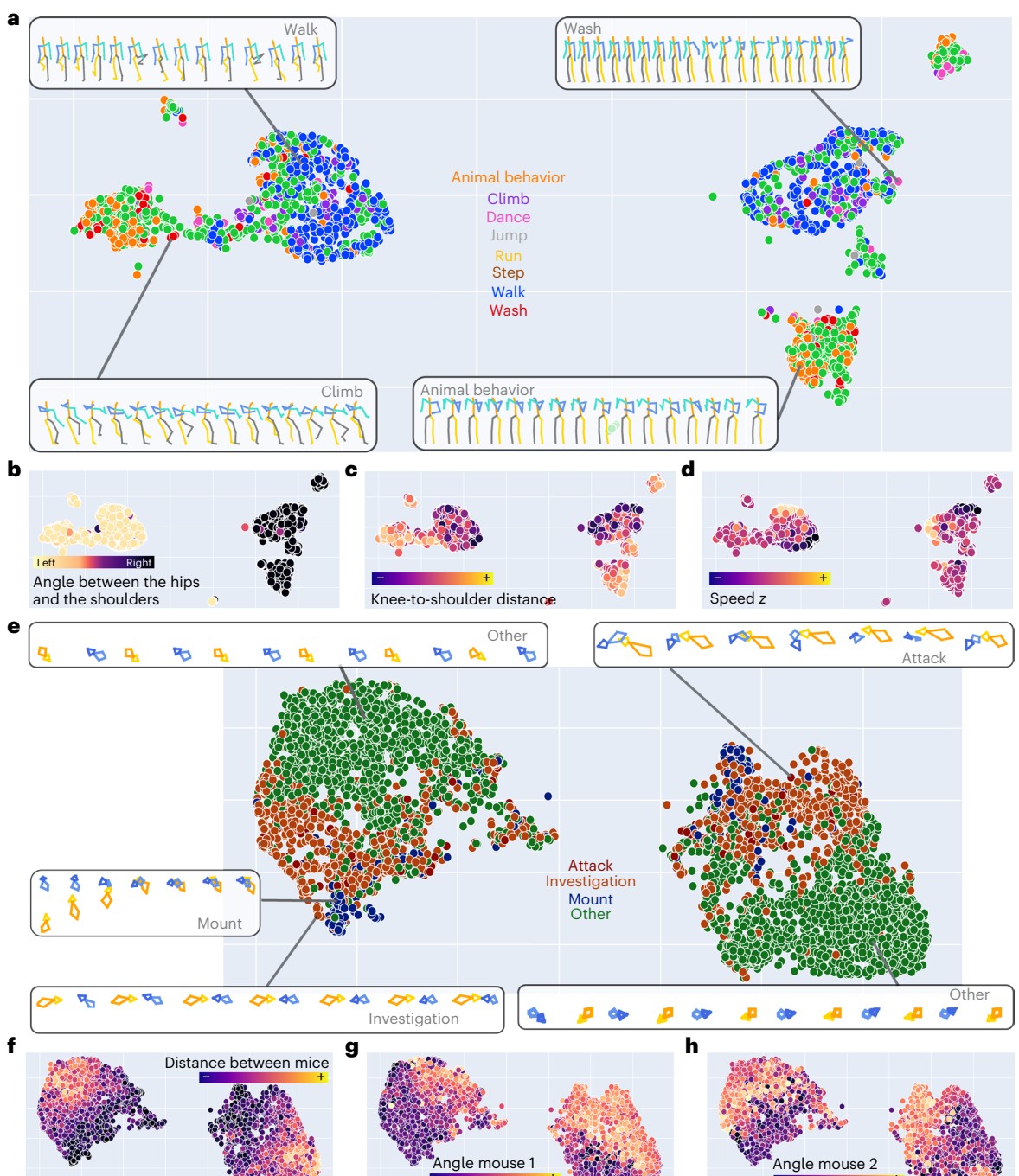

**Fig. 5 | DISK learns meaningful representations of input sequences. a–d,** UMAP plots of DISK latent space of 10-second-long sequences from the Human dataset colored by the labeled action (**a**), the angle between the hips and the shoulders— reflecting the global direction of the body (**b**), the distance between knees and shoulders (**c**) and the averaged keypoint displacement in the $z$-direction (**d**). Only sequences with an action label are displayed. One point in the plot corresponds to one sequence. Four randomly selected 3D skeleton sequences illustrating four action types are shown in **a** next to the respective data points. **e–h,** UMAP plots of DISK latent space of 2-second-long sequences from the 2-Mice-2D dataset colored according to the behavior class of the sequence (as behavior labels are available on a frame-by-frame basis, the sequence is labeled after the majority class of the contained frames) (**e**), the distance between the two mice (**f**), the angle of mouse 1 with respect to the vector basis (**g**) and the angle of mouse 2 with respect to the vector basis (**h**). One point in the plot corresponds to one sequence. 2D skeleton representations of randomly selected sequences are shown next to the plot in **e**.

learn complex, transient and context-aware correlations between keypoints and not only correlations based on a static fixed skeleton. With the fully unsupervised training scheme, further solutions for imputation of more animals separately or combined can be tested side by side and compared via the test RMSE. On the other hand, DISK is not designed to impute missing data in a dataset with only one tracked keypoint per animal (for example, centroid), such as the ones generated by idTracker[41].

DISK relies on data alone to learn dynamics and inter-keypoint dependencies. It therefore necessitates a sufficient quantity of data for training of at least 2,000 sequences based on our experience. When creating the dataset for DISK training, the length of the samples and the stride between samples can be adjusted to influence the training set size. Longer input sequences providing more context result in better performance at the cost of a longer training time. As an alternative to

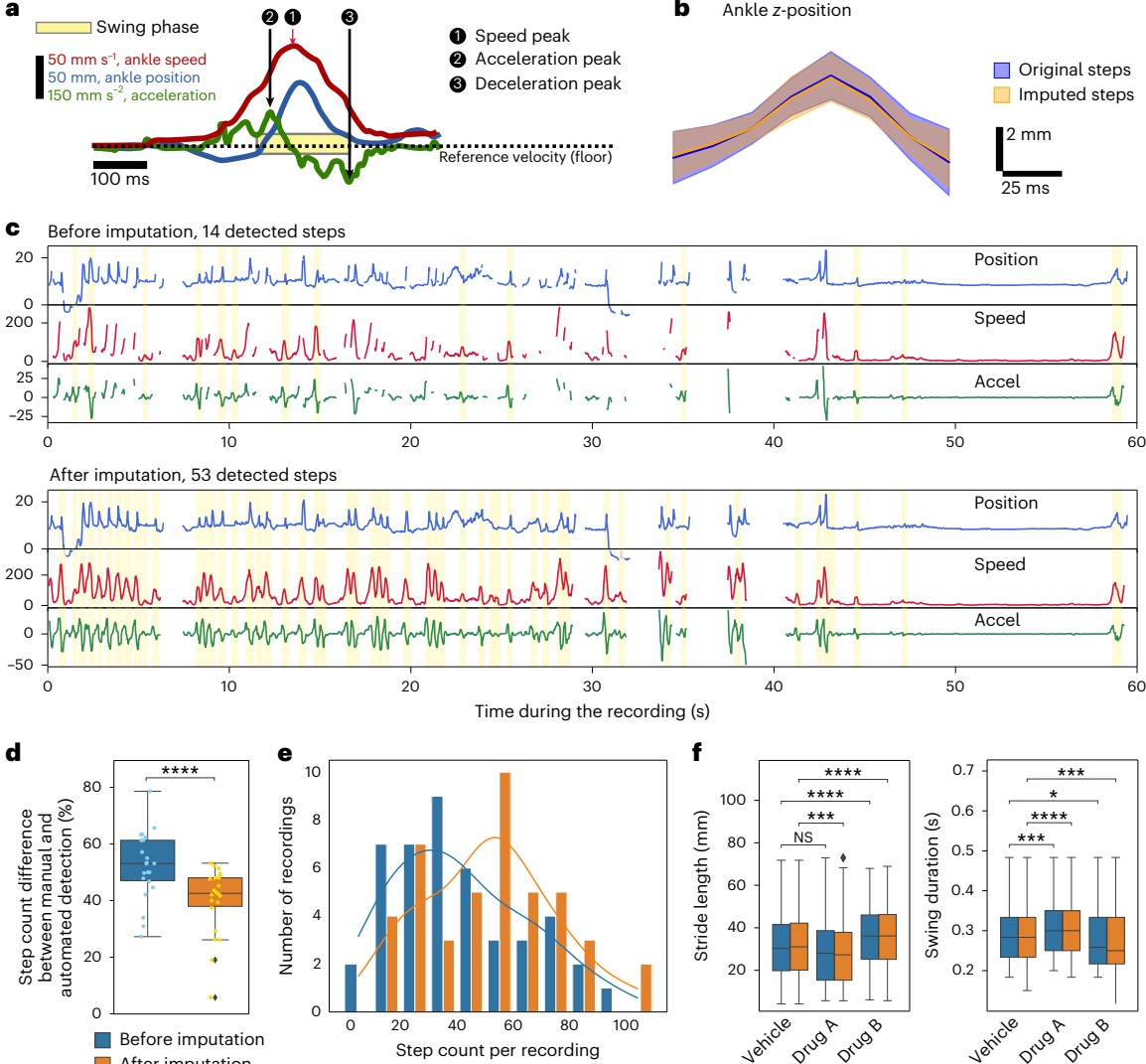

**Fig. 6 | DISK allows detection of more steps and emphasizes differences in step kinematics between different treatments. a**, Step detection principle. A swing phase of one ankle is detected as a peak on the ankle speed (1). The beginning and end of the swing phase are then defined as a peak on the acceleration before the speed peak (2), and as a valley in the acceleration after the speed peak (3). **b**, Distribution of the step ankle trajectory for original and imputed steps. The solid line represents the mean; the shaded area covers 1 s.d. above and below the mean. In orange are the steps detected on imputed data ($n = 37$), excluding the steps corresponding to the ones detected in the original data ($n = 313$). **c**, Example of a recording with detected steps before (14) versus after (53) imputation. **d**, Difference between manual step counts and automated step count before/after imputation in percent (paired two-sided $t$-test, $t = 4.894$, $P$ value $= 7.7 \times 10^{-5}$, $n = 22$ recordings). **e**, Steps per 1-min recording were counted. The histogram displays the number of recordings falling into each bin of type $[x, x + 10)$ steps. **f**, Distribution of two features of the step swing phase: stride length (total distance traveled by the ankle during the swing

phase) and duration. Two-sided $t$-tests performed on each feature comparing before and after imputation inside the same treatment are nonsignificant (number of samples per boxplot in the order of the boxplots = 313, 676, 157, 378, 552 and 1,151; independent two-sided $t$-tests on stride length: vehicle versus drug A before imputation, $P$ value $= 5.658 \times 10^{-2}$ $t = 1.911$; vehicle versus drug A after imputation, $P$ value $= 2.705 \times 10^{-4}$ $t = 3.654$; vehicle versus drug B before imputation, $P$ value $= 1.598 \times 10^{-5}$ $t = -4.339$; vehicle versus drug B after imputation, $P$ value $= 2.870 \times 10^{-9}$ $t = -5.968$; $t$-test on swing duration: vehicle versus drug A before imputation, $P$ value $= 3.952 \times 10^{-4}$ $t = -3.569$; vehicle versus drug A after imputation, $P$ value $= 1.309 \times 10^{-6}$ $t = -4.867$; vehicle versus drug B before imputation, $P$ value $= 1.960 \times 10^{-2}$ $t = 2.338$; vehicle versus drug B after imputation, $P$ value $= 4.672 \times 10^{-4}$ $t = 3.505$; NS, not significant; asterisks denote statistical significance). **d**,**f**, The box plots show the three quartiles while the whiskers extend to show the rest of the distribution, except for points that are determined to be 'outliers' using a method that is a function of the interquartile range. * $0.01 < P \le 0.05$; ** $0.001 < P \le 0.01$; *** $0.0001 < P \le 0.001$; **** $P \le 0.0001$.

the transformer backbone, GRU with the second-best overall performance also shows good results on longer sequences while being faster to train and with a lower memory requirement.

DISK allows us to take advantage of the entire behavioral experiment data, without the need to discard frames with incomplete skeleton information or to linearly interpolate the missing keypoints. In the Rat7M dataset, 44% of frames contain missing information, discarding them increases the per-frame cost of this type of data generation. Pose estimation algorithms like DeepLabCut and SLEAP can be trained to

predict the complete set of an animal's coordinates under good lighting and viewpoint conditions, and incorporating additional annotations can enhance the precision of these models. However, DISK can alleviate the annotation effort required to achieve comparable coverage and tracking accuracy. As more behavior datasets are produced and analyzed, the optimal approach to integrating these methods for accurate and efficient results will become clearer. Recovering missing data not only allows us to access more of the data but also is fundamental to allow for comparison of precise motions such as stepping or grasping.

Differences in such motions cannot be detected with general features like distance traveled or locomotion speed.

Novel recording technologies generate more high-quality data and open opportunities to record unconstrained animal behavior in challenging, natural environments. In such setups, the capability of using all recorded data is particularly important, as the exact experiments cannot be repeated. DISK allows behavioral scientists to fully exploit recordings of behavior whether generated in complex laboratory experiments or in unique natural conditions. The release of DISK as an open-source Python package aims to facilitate its adoption and support open science.

## Online content

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

## Methods

### Neural network architectures

We selected several neural network backbones for testing: transformer, GRU, TCN, ST-GCN and STS-GCN. Two of them, GRU and TCN, were designed for time-series analysis, and have been tested on skeleton data on tasks such as the estimation of 3D human pose data from monocular camera images[44,45]. GCNs have been developed specifically to handle skeleton data in a different task to ours, namely action recognition. Transformers are currently the state-of-art models in sequential data-related tasks and show the highest capacity to learn complex and long-term patterns in sequences[34,46,47].

**GRU.** We devised a bidirectional GRU-based architecture for the task of missing data imputation in time-series pose data. The model comprises *n* GRU layers followed by an output linear layer. We tested several hidden layer sizes from 32 to 512, one to three recurrent layers, and two values of dropout, 0 and 0.2, applied before the output linear layer. GRUs with three layers, hidden size of 512 and no dropout showed the best performance. Our GRU architecture follows a previously published model, BRITS[12], with two modifications: (1) we replaced the long–short-term memory (LSTM) layers with GRU layers; (2) we added an output linear layer. Replacing LSTM layers with GRU layers was motivated by the observation that GRUs perform equally well or better than LSTMs on small datasets[48,49]. Adding a linear output layer did not affect the performance of GRUs but allowed us to adapt the size of the network to different datasets and numbers of keypoints by decoupling the hidden and the output size. We chose a bidirectional flow to leverage information from before as well as after the gap.

**TCN.** We tested a simple version of a TCN with causal one-dimensional (1D) convolutions with respective dilation rates of 1, 2, 4 and 8 for the subsequent layers as in the original WaveNet paper[50]. We used four residual blocks in the TCN. A block is composed of two convolutional layers with ReLU activation and 0.2 dropout. All the hidden layers have 256 units.

**GCN.** GCNs received much attention in the context of handling skeleton data for action recognition[51–53]. These networks use graph representation of the relationships between input keypoints. We tested two GCN architectures for the task of keypoint imputation: ST-GCN and STS-GCN[51,52]. We parametrized ST-GCN to five layers with a hidden size of the first layer of 64, which decreased by a factor of 2 at each following layer. We chose a smaller hidden size for this network compared to other tested networks because this architecture does not scale well with increasing hidden size (Supplementary Table 1). Indeed, with a larger hidden size, the network did not fit in one GPU V100 32-GB core memory, which is impractical for the deployment of the method. STS-GCN was used with two graph encoding layers and two temporal convolutional decoder layers, with a hidden size of 256 and a dropout of 0.2. ST-GCN takes as input the skeleton while STS-GCN builds an adjacency matrix during training.

**Transformer encoder.** Transformer architectures, and mixed architectures with graph and transformer components, are used in action recognition tasks[54–57]. Our architecture was inspired by refs. 58,59. Instead of performing a linear projection of one pose with all the keypoints into a single vector, we flatten the posture and apply the same learned linear projection layer separately on each keypoint. In the first scenario masking can only be done at a pose level and does not allow the masking of individual keypoints. In the second scenario after flattening, masking of an individual keypoint is possible and allows the network to exploit the relationships between keypoints for imputation. Parallel to the linear projection of the coordinates of each keypoint, the time information (1D vector of time points), the keypoint identity and the missing mask are each passed through an Embedding layer, which acts

like a look-up table (see PyTorch Embedding module). We provide the information about the missing coordinates using a binary mask indicating which values are missing. In the case of transformers, we use the binary mask two ways: the binary mask is concatenated to the input coordinates (similar to other architectures; see 'Artificial gap insertion'), and the binary mask is embedded via an Embedding layer. The time embedding, keypoint identity embedding, binary mask embedding and the linear projection are summed into the input tensor, which is fed to the transformer encoder layers. Based on our experiments, we found that four transformer layers with an internal dimensionality of 128, a model dimension of 128, and eight attention heads as well as layer normalization and a small batch size (≤32) gave the best results across datasets. A linear layer was used to produce the final output. When testing the transformer architecture with different input lengths (Extended Data Fig. 5) on a V100 GPU with 32-GB RAM, the longest sequence we could evaluate was a length of 240, because a length of 480 with six keypoints (2-Fish dataset) exceeded the available memory.

### Datasets

We tested the different neural network solutions on seven kinematics datasets: two 3D motion capture mouse datasets (FL2, in which mice perform floor exploration; and CLB, in which mice climb on the outside of a mesh wheel[9]), a 2D dataset with two mice (2-Mice-2D), a 3D markerless *Drosophila* dataset (DF3D)[42], a 3D Human motion CMU motion capture dataset (Human)[60], a 3D motion capture rat dataset (Rat7M)[4] and a 3D markerless dataset with two Zebrafish (2-Fish)[31].

The 3D mouse datasets comprise 1-min recordings of 22 mice under pharmacological interventions affecting their behavior[9]. Each mouse is recorded freely behaving in two different setups: an open-field exploration (mouse FL2 dataset, 101 recordings) and a climbing vertical mesh (mouse CLB dataset, 102 recordings). The order of the recordings (setup and drugs) is randomized among mice. The Qualisys motion capture system outputs coordinates for eight keypoints in 3D at a frequency of 300 Hz, which we downsampled to 60 Hz.

The 2-Mice-2D dataset corresponds to the task 1 of the CalMS21 CVPR 2021 challenge[39] accessible on the AICrowd website (https://www.aicrowd.com/challenges/multi-agent-behavior-representation-modeling-measurement-and-applications/problems/mabe-task-1-classical-classification/). Keypoint trajectories from top-down views of two mice under a resident–intruder scheme are available. We only considered the train split from the challenge containing 70 recordings from 3.2 s to 12 min at 30 Hz. Frame-by-frame annotations of four possible behaviors (attack, investigate, mount or other) are available.

The DF3D dataset from ref. 42 consists of videos of tethered flies with different genetic backgrounds. Immobilized at the thorax, their limbs can move a floating ball. This setup generates a very close and precise view of the animal with seven synchronized cameras. The tethering restricts possible behaviors to a few, such as walking forward and backward, grooming and resting. In total, 199 videos of 9 s each at a recording frequency of 100 Hz are available. Thirty-eight keypoints on the legs, body and head are tracked via pose estimation algorithms on the 2D videos and their 3D positions have been reconstructed in previous work[42].

We used the subset of the CMU Human MoCap dataset (http://mocap.cs.cmu.edu/) curated by ref. 60 and accessed at https://ericguo5513.github.io/action-to-motion/#data/. Human actors perform different actions including running, walking, jumping and climbing. This dataset contains 1,577 motion sequences of more than 60 frames with 20 3D keypoints at a frequency of 12 Hz. The original units of the CMU Human MoCap dataset are in inches multiplied by 0.45 (https://mocap.cs.cmu.edu/faqs.php).

The Rat7M dataset consists of the 3D motion capture recordings[4], and is available online (https://github.com/spoonsso/dannce/). Freely behaving rats are recorded in a transparent round arena at 30 Hz with six synchronized cameras. The seven recordings depict five different animals, and have a length between 33 min and 2 h 10 min.

The 2-Fish dataset consists of 22 videos of six pairs of adult male Zebrafish recorded during 90 min or longer with three synchronized cameras (two side views and one top-down view) at 100 Hz. Three body points (head, pectoral fin and tail) are tracked with SLEAP[2] on the top-down view before animal ID tracking with idTracker.ai (https://idtracker.ai/latest/) and 3D view reconstruction. The recordings start with the introduction in the fish tank of two foreign fish. Only recordings displaying rigorous interaction (maneuvers, chase and fight) were kept by the original authors[31].

## Data preparation

Input data are composed of 3D coordinates of body keypoints. Original data files are loaded and split into train, test and validation sets. Data loaders are available for a variety of formats, including the ones from the tested dataset, DeepLabCut and SLEAP outputs. In all the datasets except Rat7M, entire recordings were separated between the train, test and validation sets: a given recording was not partitioned among the sets but assigned entirely to one of the sets. In the Rat7M data, only seven 3D recordings from five animals were available. The lengths of the recordings were very variable, from a few minutes to several hours. Separating these animals' recordings into train, test and validation sets would result in highly unbalanced sets. Instead, we split each sequence in 70%/15%/15% of the frames. The starting 70% were assigned to the train set, the next 15% to the validation set and the final 15% to the test set. With this partition, there is no overlap between sets and the train set contains only prior information to the test or validation sets.

Once assigned to sets, the recordings were split into 60 frame-long sequences with a stride depending on the dataset (note the different frequencies used for each dataset in Extended Data Table 1). The stride—the shift between two consecutive sequences—was set to 30 (half a sample), except for the smallest dataset, DF3D, where the stride was set to 5 to obtain a larger number of sequences; for 2-Mice-2D, where the stride was set to 60; and for the large 2-Fish dataset, where the stride was set to 120 to decrease the number of sequences. We will refer to these 60 frame-long sequences as samples. These samples are direct inputs to the neural networks. The number of samples in each dataset is reported in Extended Data Table 1.

Only complete samples without any missing values were kept for training and evaluation. We tested the approach developed in the SAITS paper[13]. All available samples—even those with missing data— are included in the training set, and additional gaps are introduced on top of the existing ones. SAITS loss consists of two parts: one part evaluating the reconstruction on non-missing coordinates, and one part evaluating the imputation on the artificial gap. Although this approach increases the number of available samples for training, it did not improve the imputation in our experiments.

## Input sequence normalization

We applied two transformations to the input samples: a ViewInvariant rotation, and a normalization[61,62]. The ViewInvariant transformation rotates a sequence so that, in the middle frame of the input sample, the body always faces the same direction. While not removing all variability, this rotation prevents the neural network from learning the position or rotation of the body in the sample and brings its focus to the intrinsic motion dynamic. In practice the body barycenter and the dominant vector in the $x$–$y$ plane in the middle frame of the sequence were used to compute the rotation aligning this dominant $x$–$y$ vector with the $(x = 1, y = 0)$ vector. This rotation was applied to all coordinates of the entire sample. We did not apply a rotation in the $z$ axis. Subsequently a min–max normalization was applied to all the coordinates of the sequence bringing every coordinate to the −1 and 1 range.

## Artificial gap insertion

FL2, CLB and Rat7M datasets contain natively missing values. Different keypoints can have different probabilities of being missing, and certain gap lengths can be more probable than others. These characteristics of the missing process are dataset dependent, and we mimicked them in gap generation during training. In practice, three probability distributions are estimated: the probability of a keypoint $k$ being missing $\hat{P_k}$, for a given keypoint, the probability of a gap length $n$, $\hat{P}(\text{length} = n|k)$, and the probability of inter-gap length $\hat{P}(\text{length} = n|\text{inter})$. During the training, for each selected sample, an artificial gap is inserted by drawing randomly a keypoint according to the approximated $\{\hat{P_k}\}_k$, then by drawing randomly the length of the artificial gap according to $\hat{P}(\text{length} = n|k)$, and finally by drawing randomly the starting position of the gap according to $\hat{P}(\text{length} = n|\text{inter})$.

In the DF3D, 2-Mice-2D or Human datasets, no coordinates were indicated as missing by a specific value or as potentially missing by a confidence score. In these datasets, we used a uniform distribution for $\hat{P}(\text{length} = n|k)$. In Human and 2-Mice-2D datasets, we also assumed a uniform probability for $\hat{P_k}$. In the DF3D dataset, the $\hat{P_k}$ of 20 keypoints close to the fixed thorax of the fly were set to 0, as they were not displaying much position change.

We restricted the possible gap lengths to the interval [1, 58] in our samples. By capping the gap length to 58, we leave the first and last time point of the missing keypoint, allowing for comparison of imputation accuracy with linear interpolation.

Missing values were replaced with zeros—a value within the range of possible coordinate values. Therefore, we used an additional binary mask of the same length as the input sequence indicating with 1 s where the missing values are. For all backbones except transformer, this mask is concatenated as another dimension to the input sequence, so each keypoint has four associated values ($x$, $y$, $z$, missing) per time frame with 'missing' being 0 or 1.

## Switch data augmentation module

We added to DISK code a switch data augmentation module, which at training time randomly switches two keypoints for a given sub-sequence of the input with a probability set by the user. For all datasets, we used a probability of 0.1 at training time. During training, the loss is computed on the non-switched version of the input sequence. Our goal was to make DISK more robust to potential switches happening in the real data. As detecting real switches in the data remains very challenging, we evaluated this technique at test time on artificially switched keypoints; for this step the probability was set to 0.5 to compare the performance on samples containing, or not, switches. Results are displayed in Extended Data Figs. 5 and 6 and Supplementary Fig. 3.

## Training procedure

We used L1 loss between the ground truth and the reconstructed gap sequence, that is, excluding the positions and keypoints outside the introduced gap, for all models but the ones with a probabilistic head. For models with a probabilistic head, we used negative log likelihood[63]. Each training lasted for 1,500 epochs. The network at the epoch with the lowest RMSE (see 'Metrics') on the validation set was saved and evaluated on the test set. All RMSEs reported in the results are calculated on the test set. We used the Adam optimizer and a learning rate scheduler with a rate of 0.95 and steps equal to 500. For all networks the starting learning rate was 0.001, except for GRU, which was 0.0001, as a higher learning rate showed spikes and instability in the loss during training. All the code used in this work was written in Python and PyTorch and is available at https://github.com/bozeklab/DISK.git/.

## Metrics

We implemented three metrics: RMSE, MPJPE and PCK. All metrics are only calculated on keypoints in frames that are missing.

RMSE is defined as:

$$\text{RMSE} = \sqrt{\frac{1}{N} \sum_{k,t} \left[ (\hat{x}_{k,t} - x_{k,t}^{\text{GT}})^2 + (\hat{y}_{k,t} - y_{k,t}^{\text{GT}})^2 + (\hat{z}_{k,t} - z_{k,t}^{\text{GT}})^2 \right]},$$

with $\hat{x}_{k,t}$ the imputed value for keypoint $k$ at time $t$ ($\hat{y}_{k,t}$ and $\hat{z}_{k,t}$) and $x^{GT}_{k,t}$ the ground-truth value ($y^{GT}_{k,t}$ and $z^{GT}_{k,t}$). We only considered where the keypoints are missing and normalized the sum by counting the number $N$ of missing positions across keypoints. For 2D, we used the same formula ignoring the $z$ term.

MPJPE is defined as:

$$\text{MPJPE} = \frac{1}{N} \sum_{k,t} d_t(\hat{k}, k^{GT}),$$

with $d_t$ being the Euclidean distance (in 2D or 3D) between imputed keypoint $\hat{k}$ and ground-truth keypoint $k$ at timestep $t$ defined in 3D as $\sqrt{[(\hat{x}_{k,t} - x^{GT}_{k,t})^2 + (\hat{y}_{k,t} - y^{GT}_{k,t})^2 + (\hat{z}_{k,t} - z^{GT}_{k,t})^2]}$.

PCK is defined as:

$$\text{PCK@th} = \frac{\sum_{k,t} \mathbb{1}_{d_t(\hat{k}, k^{GT}) < \text{th} \times \text{max\_dist}}}{N},$$

with 'th' a number between 0 and 1, and 'max_dist' as the maximum distance between two keypoints in a frame for the whole dataset[8]. We chose a very low threshold, th = 0.01, which corresponds to 1% of the maximum inter-keypoint distance; other studies usually choose th = 0.1.

## Computation of movement and periodicity per sample

For each sample, the movement is defined as the averaged absolute difference between each coordinate and its previous time frame. The periodicity is defined as the highest weight corresponding to one frequency in the 1D Fourier spectrum of each coordinate ($x$, $y$ or $z$) over all keypoints. We used the implementation of fast fourier transform from 'scipy.signal'. These two quantities are computed on original sequences without missing data.

## Comparison with other methods

For Keypoint-MoSeq and OptiPose, the original implementation was used. For MarkerBasedImputation, we reimplemented the code as the original codebase was not running on our setup due to old software package dependencies.

At training time, the different methods (Keypoint-Moseq, Opti-Pose, MarkerBasedImputation, DISK) were provided with the same datasets (training data without keypoints missing; Extended Data Table 1).

For Keypoint-MoSeq, the number of latent dimensions was set to the number of dimensions representing more than 90% of the variance, as described in the method's tutorial. Following the online example, we kept the default values for the number of initial iterations of the autoregressive model (50) and for the additional iterations (500). For OptiPose, we kept the parameters as given in the code base (num_samples = 20,000, batch_size = 100, max_epochs = 1,500). We used a one-size-fits-all approach like for the other methods, and chose ten as the number of parallel context models (CMs), ten as the number of sub-CMs and four as the number for the multi-head parameter of the self-attention layer. However, we decreased the learning rate to $1 \times 10^{-4}$ or $5 \times 10^{-5}$, as the suggested value $3.1 \times 10^{-4}$ was resulting in loss explosion. For MarkerBasedImputation, the preprocessing involving centering, rotating and $z$-scoring the input sample was done on already masked segments to avoid leakage from the ground truth to the input sequences. We kept the hyperparameters the same as in the code base, with constant-sized models and training ten models for 30 epochs for the ensemble prediction.

Metrics and plots were calculated on the held-out test dataset. The outputs of each method were saved in the original coordinate space on which the metrics (PCK, RMSE, MPJPE) were then computed. At test time, to compare the performance of the models on exactly the same sequences, sequences of 60 frames were sampled from the test dataset

and random gaps were added with the DISK test script. Additionally, we ran all three methods on 1-min recordings from the mouse FL2 test dataset (Fig. 2h,i).

The necessary conda environment specifications and additional scripts are available in an additional public code database at https://github.com/bozeklab/DISK_paper_code/.

## Latent projection and behavior classification

For the Human and the 2-Mice-2D datasets, behavior class labels are available. For the 2-Mice-2D dataset, the labels are available in a frame-by-frame manner. We then take the majority class of the frames to assign a label to a sequence. For the Human dataset, behavior class labels are given per recording, so we assign the recording label to each of its sub-sequence. For the Human dataset, we removed the samples labeled as unknown before the classification.

We used the output of the encoder layers before the final linear layer for the UMAP plots. We computed the angle between hips and shoulders, the hip-to-shoulder distance, the knee-to-shoulder distance, the speed in the $x$–$y$ plane, the distance between two mice, angle mouse 1 and angle mouse 2, and the speed in the $z$-plane on the input coordinates without missing data. We computed the UMAP on samples from the training set using default parameters of the umap-learn Python package, and projected samples from training validation and test sets.

We used a random forest classifier with default parameters of the scikit-learn Python package and fit on the latent vectors extracted the same way as for UMAP. We trained the random forest on the same training set as DISK and reported prediction results on the test set.

## Step detection and kinematics

We followed the definitions from ref. 9 and used resampled keypoint trajectories at 60 Hz from the original 300-Hz recordings to do the imputation, and compute the locomotion features before and after imputation. For the imputation, a DISK-proba model was trained on multiple missing keypoints with inferred probabilities. As ankles were dropped in the mouse FL2 dataset, we pulled more data from an additional experiment to gather a sufficient data amount for the training. This additional experiment corresponds to the same open floor exploration task with different pharmacological treatments and slightly different keypoints. We selected eight common keypoints: right ankle, left ankle, right knee, left knee, right hip, left hip, right back, left back. As for the mouse FL2 dataset, original data were downsampled to 60 Hz with a sample length of 60 and a stride of 30. After inspection of imputations on the held-out test dataset, we rejected imputed samples when the estimated error was greater than 0.1 for the imputation on the entire dataset.

For the step kinematics, we computed the instantaneous ankle speed (the two ankles are considered separately) and smoothed it over 0.1 s. We detected peaks on the smoothed ankle speed with a minimum height of 20, a minimum prominence of 21, and a duration in the range from 0.16 s to 2 s. For each detected speed peak, we considered potential start and end points as the closest valleys in the ankle acceleration. If the start or end points were not found, then the peak was discarded and the sequence was not considered a step. Steps whose duration was lower than 0.6 s, and steps where the ankle position was not lower at the start and the stop compared to the swing phase, were discarded as well. Manual counts on 22 recordings were provided by B.M.I.-J. The swing phase features were computed as follows[9]:

- Stride length: length of the 3D trajectory of the ankle or knee marker between the frames defined as swing start and swing end
- Swing duration: time difference between detected start and stop of the swing phase (in seconds)

## Reporting summary

Further information on research design is available in the Nature

Portfolio Reporting Summary linked to this article.

## Data availability

The following original datasets used in the publication are available online: Fish (https://doi.org/10.5281/zenodo.10103746; ref. 64), MABe (resources section of https://www.aicrowd.com/challenges/multi-agent-behavior-representation-modeling-measurement-and-applications/problems/mabe-task-1-classical-classification/), DF3D (https://dataverse.harvard.edu/dataverse/DeepFly3D/), Human (CMU Mocap section at https://ericguo5513.github.io/action-to-motion//#data) and Rat7M (https://figshare.com/collections/Rat_7M/5295370/3; ref. 65). The mouse data are available upon request from the original authors[9]. Data from the same recording rig are available at https://doi.org/10.5281/zenodo.15493338 (ref. 66). Additionally, to support the DISK method, we provide processed datasets at https://doi.org/10.5281/zenodo.15800034 (ref. 67).

## Code availability

The DISK method is available in GitHub at https://github.com/bozeklab/DISK.git/ under an MIT License. A frozen version is available at https://github.com/bozeklab/DISK/releases/tag/v1.0.0/. Trained model checkpoints are available at https://doi.org/10.5281/zenodo.15800034 (ref. 67). Specific scripts and data files to reproduce the results and figures in this publication are available at https://github.com/bozeklab/DISK_paper_code under an MIT License. All the custom code was developed using Python 3.9 and DISK v1.0.0. The list of packages and versions are available in each code base on GitHub (https://github.com/bozeklab/DISK/blob/main/DISK/requirements.txt; https://github.com/bozeklab/DISK_paper_code/blob/main/requirements.txt) in the 'requirements.txt' file.

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

## Acknowledgements

K.B. and F.R. were funded by the BMBF program for Female Junior Researchers in Artificial Intelligence 01IS20054. F.R. received funding from North Rhine-Westphalia Ministry of Culture and Science, KI-Starter 2210kis004. T.B. and M.M. were funded by the Köln Fortune program. T.D.P. was funded by the National Institutes of Health (NIH; 1RF1MH132653-01). This research was supported by Japan Society for Promotion of Science (JSPS) Fellowship for Overseas Researchers (P17388), Kakenhi Grant-in-Aid for JSPS Fellows (17F17388) and Kakenhi Grant for Scientific Research (21K06399) awarded to B.M.I.-J. G.J.S. and L.O. were supported by grant no. RGP0055 from the Human Frontiers Science Program and by funds from Vrije Universiteit Amsterdam. G.J.S. was additionally supported

by OIST Graduate University. M.Y.U. received no specific funding for this work. The funders had no role in study design, data collection and analysis, decision to publish or preparation of the manuscript. We are grateful for the help and support provided by the Animal Resources Section (ARS) of Core Facilities at Okinawa Institute of Science and Technology Graduate University, as well as the entire Neuronal Rhythms in Movement (nRIM) unit at Okinawa Institute of Science and Technology (OIST) for helpful discussions. The Human data used in this project were originally obtained from https://mocap.cs.cmu. edu/. The database was created with funding from National Science Foundation EIA-0196217.

## Author contributions

F.R and K.B. designed the overall study. F.R., K.B. and T.D.P. designed the benchmark experiments (metrics, switch). L.O. and G.J.S. provided the fish data, and together with F.R. and K.B. specifically analyzed the fish data. B.M.I.-J. and M.Y.U. provided the motion capture mouse data, and together with F.R. and K.B. proposed the step analysis. F.R., M.M. and T.B. wrote and tested the code, and generated results. F.R. generated the figures. F.R. and K.B. wrote the paper. All authors read and approved the final paper.

## Funding

## Competing interests

The authors declare no competing interests.

## Additional information

**Extended data** is available for this paper at https://doi.org/10.1038/s41592-025-02893-y.

**Correspondence and requests for materials** should be addressed to France Rose or Katarzyna Bozek.

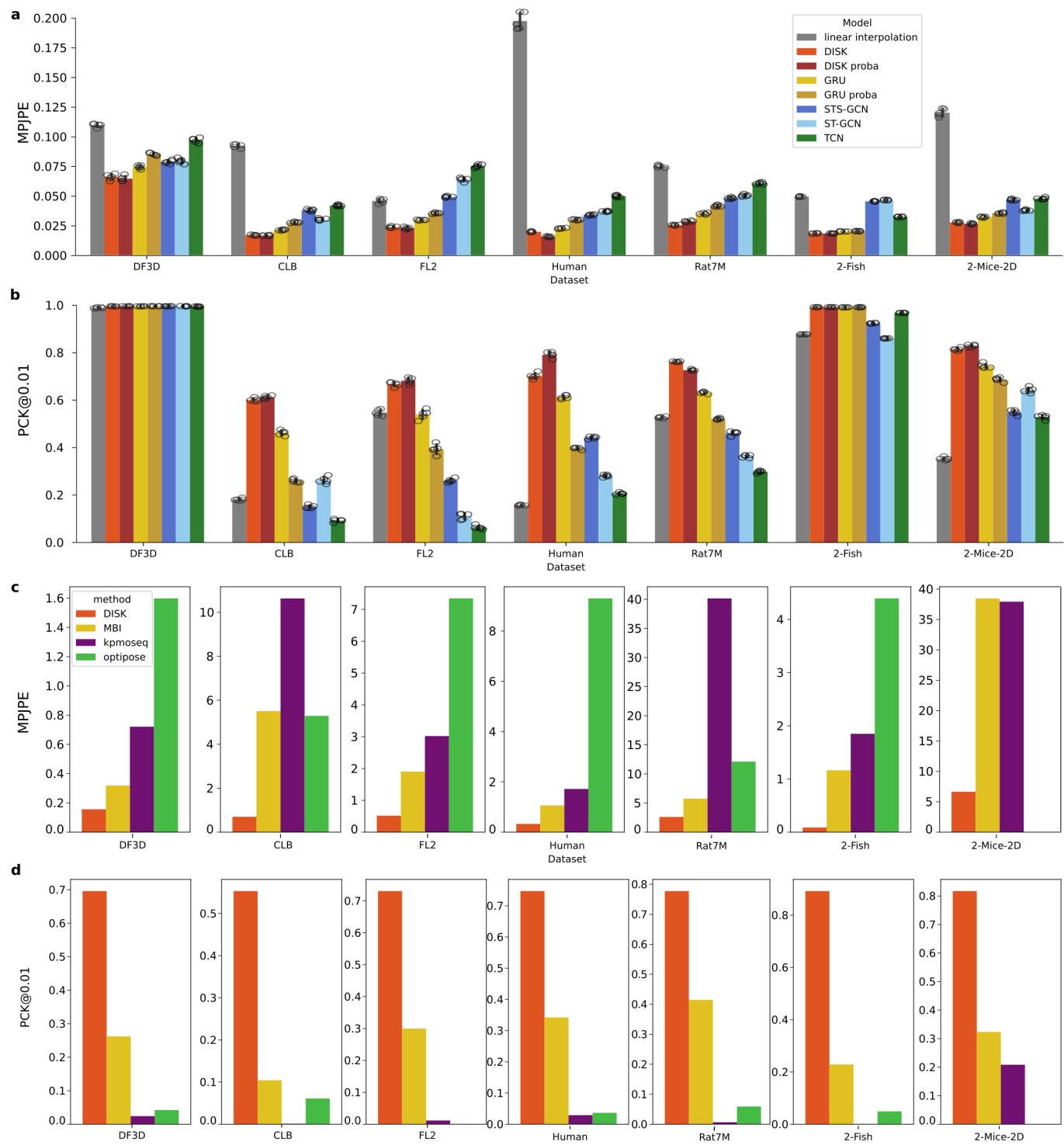

**Extended Data Fig. 1 | Comparison of different architectures and methods.**
Comparison of different architectures according to the MPJPE (**a**) (normalized units) and PCK@0.01 (**b**) metrics, and of different methods according to the MPJPE (**c**) (original units, millimeters for all datasets except Human) and PCK@0.01 (**d**) metrics. Bar plots in **a** and **b** represent the mean of five test runs of the same model, and error bars the standard deviation. Bar plots in **c** and **d** represent the mean of $n = 3,392$ gaps for DF3D, $n = 6,116$ for CLB, $n = 9,120$ for FL2, $n = 9,328$ for Human, $n = 26,532$ for Rat7M, $n = 13,008$ for 2-Fish and $n = 74,324$ for 2-Mice-2D.

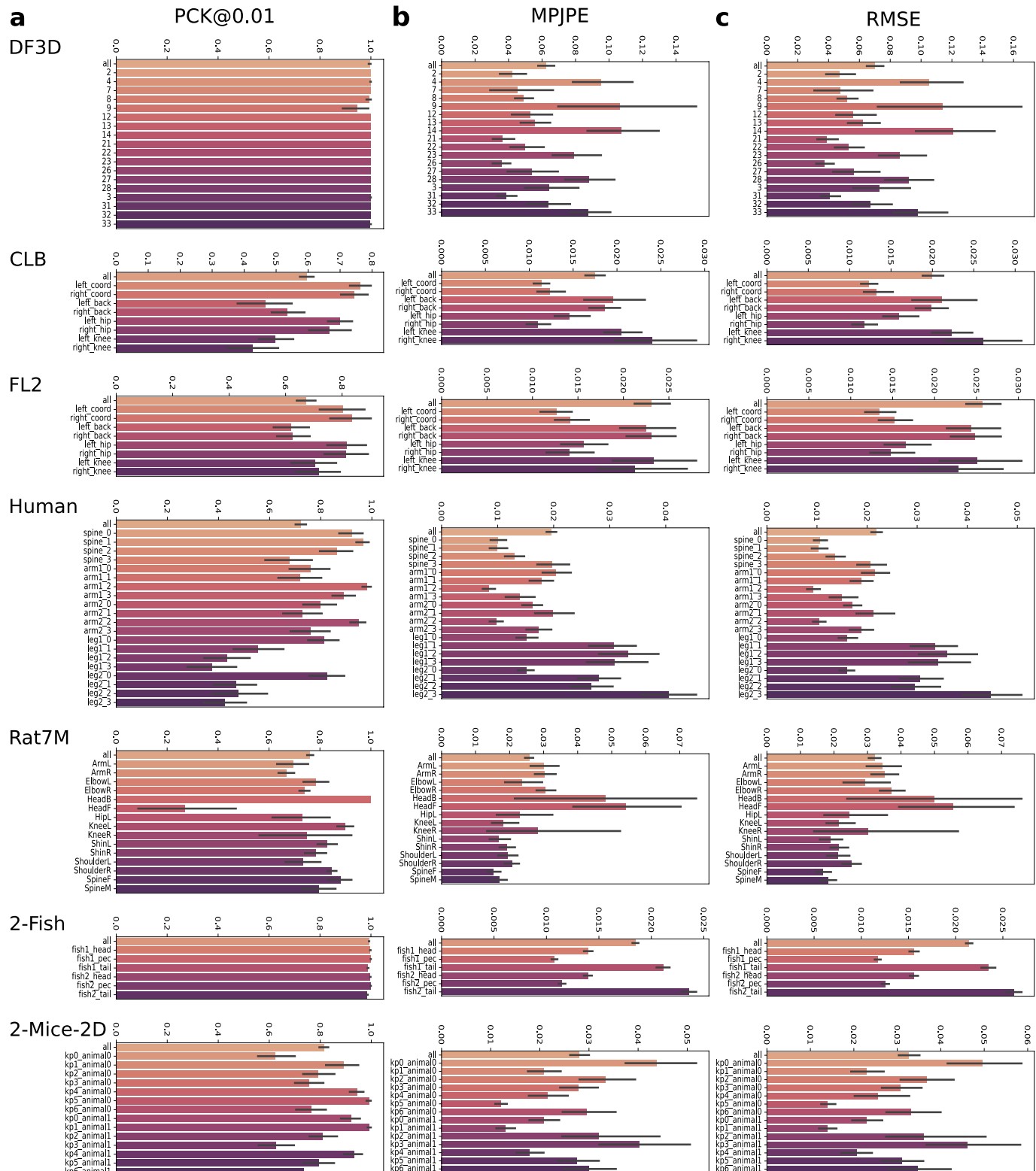

**Extended Data Fig. 2 | Metrics per keypoint and average across keypoints.**
PCK@0.01 (**a**), RMSE (**b**), and MPJPE (**c**) per keypoint and average across
keypoints ('all'; in original units, millimeters except for the Human dataset) in
datasets: DF3D (keypoints numbered as in the original data https://github.com/
NeLy-EPFL/DeepFly3D), FL2, CLB, Human, Rat7M, 2-Fish, 2-Mice-2D (kp0, nose;
kp1, left ear; kp2, right ear; kp3, neck; kp4, left abdomen; kp5, right abdomen;
kp6, tail base). Barplots represent the average and error bars the 95% confidence
interval of number of gaps.

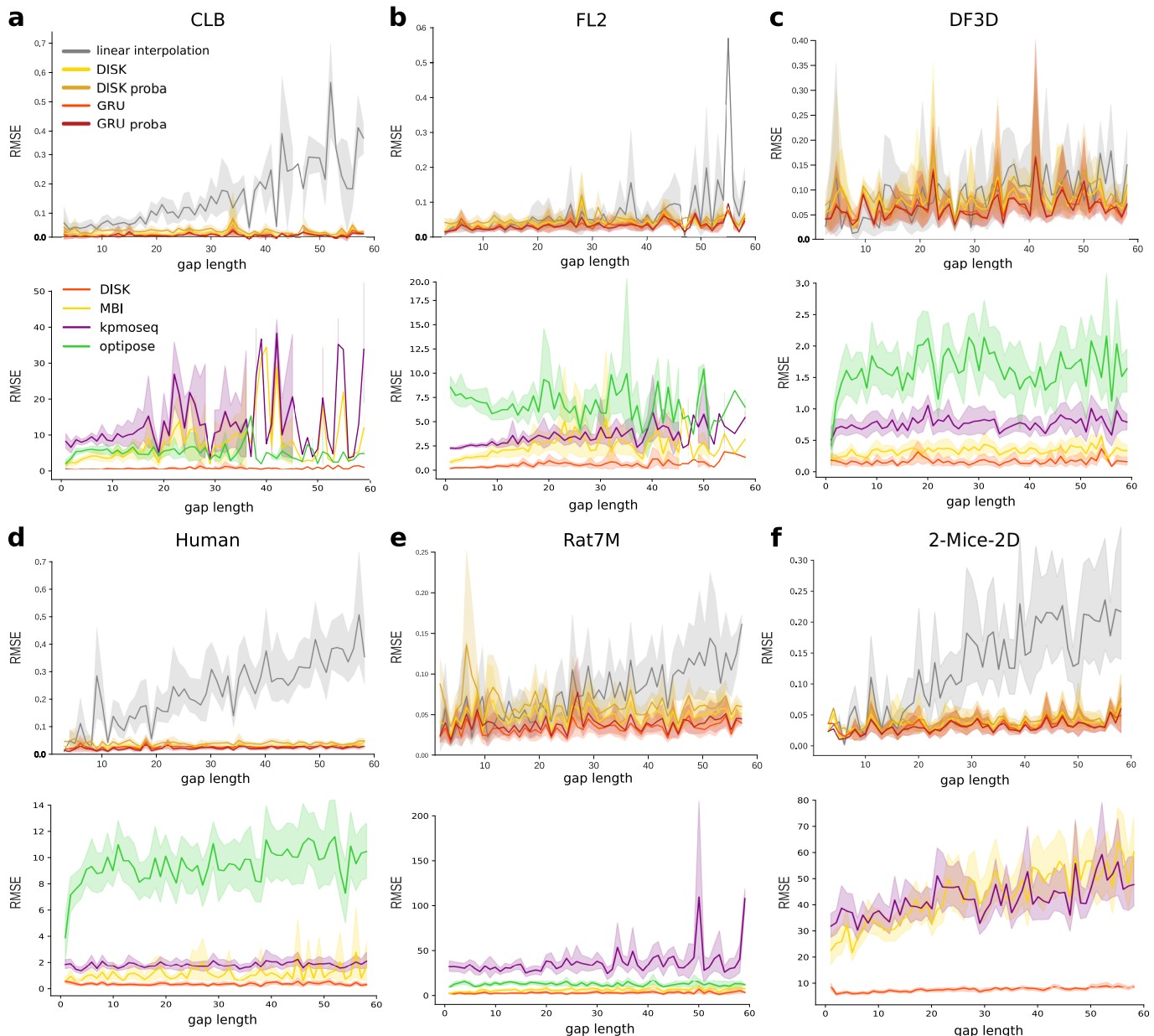

**Extended Data Fig. 3 | RMSE with respect to the gap length for the other datasets.** Panels in the first row compare the different DL architectures and report RMSE in normalized units. Panels in the second row compare the different methods and report RMSE in the original coordinate units (millimeters for all datasets except the Human dataset). **a** CLB, **b** FL2, **c** DF3D, **d** Human, **e** Rat7M, **f** 2-Mice-2D. The line represents the mean and the shaded area a 95% confidence interval.

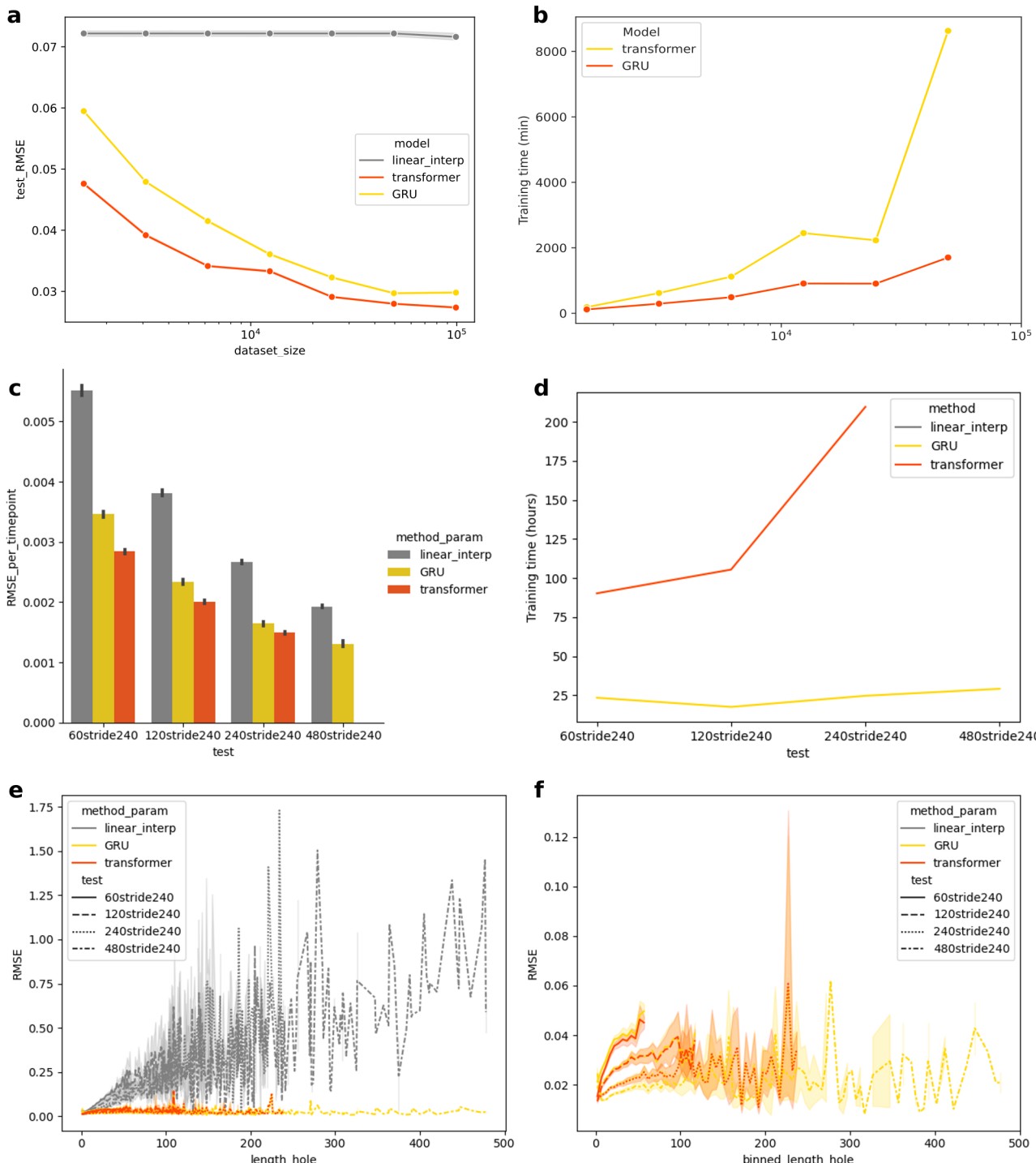

**Extended Data Fig. 4 | DISK performance dependence on dataset size and input length (2-Fish dataset). a**, Performance when decreasing the size of the dataset by subsampling. **b**, Training time for different sizes of subsampled datasets (batch size = 32) for GRU and transformer (DISK). **c**, RMSE for GRU, transformer (DISK), and linear interpolation. Bar plots represent the mean and standard deviation of test runs of the same model. **d**, Training size for different input length (dataset size kept constant). The evaluation for length 480 for the transformer architecture is missing as one sample did not fit in the memory of one V100 GPU. **e**,**f**, RMSE with respect to gap length. The shaded area corresponds to the 95% confidence interval. **f** is a zoom of **e** without the linear interpolation line.

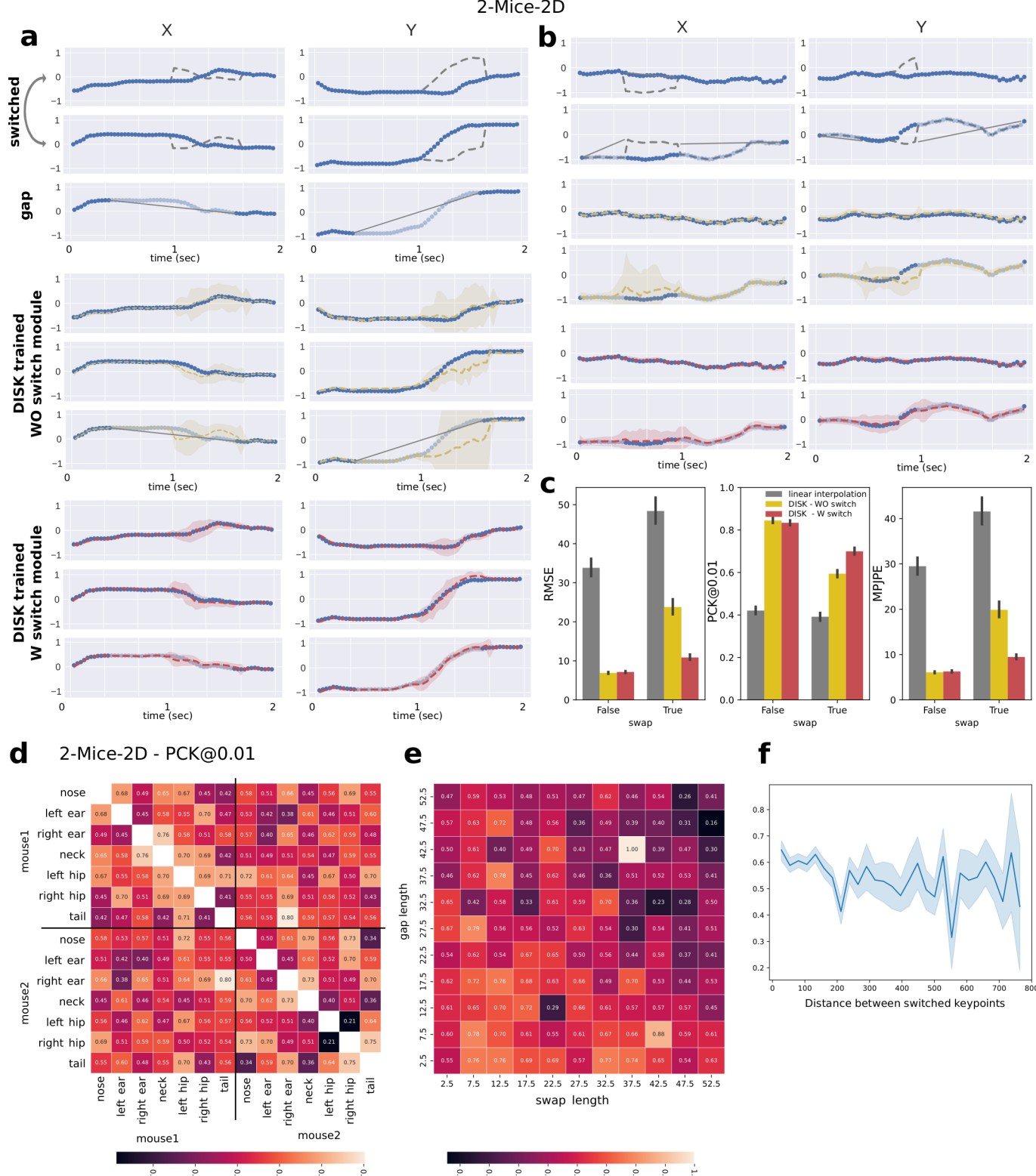

**Extended Data Fig. 5 | Imputation of switched keypoints. a**,**b**, Examples with switched keypoints at test time from the 2-Mice-2D dataset. The blue trajectory represents the ground truth; the dashed gray line is the sample with the switched keypoints given as input to the DISK-proba models; the straight gray line is the linear interpolation of the gap. The trajectory and uncertainty are displayed in yellow for the DISK-proba model trained without switch data augmentation and in red for the DISK-proba model trained with switch data augmentation. **c**, Averaged RMSE, PCK@0.01 and MPJPE for linear interpolation and the two DISK-proba models expressed for test samples with (W switch) and without switched keypoints (WO switch). The bar plots represent the mean and the error bars the 95% confidence interval. **d**–**f**, Averaged PCK@0.01 on the samples containing switched keypoints for the 2-Mice-2D dataset as a function of different parameters. All RMSEs and MPJPEs are expressed in mm. **d**, PCK@0.01 as a function of pairs of switched keypoints. **e**, PCK@0.01 as a function of gap length and switch length (expressed in frames). **f**, PCK@0.01 as a function of the averaged distance between switched keypoints. The line represents the mean and shaded area the 95% confidence interval.

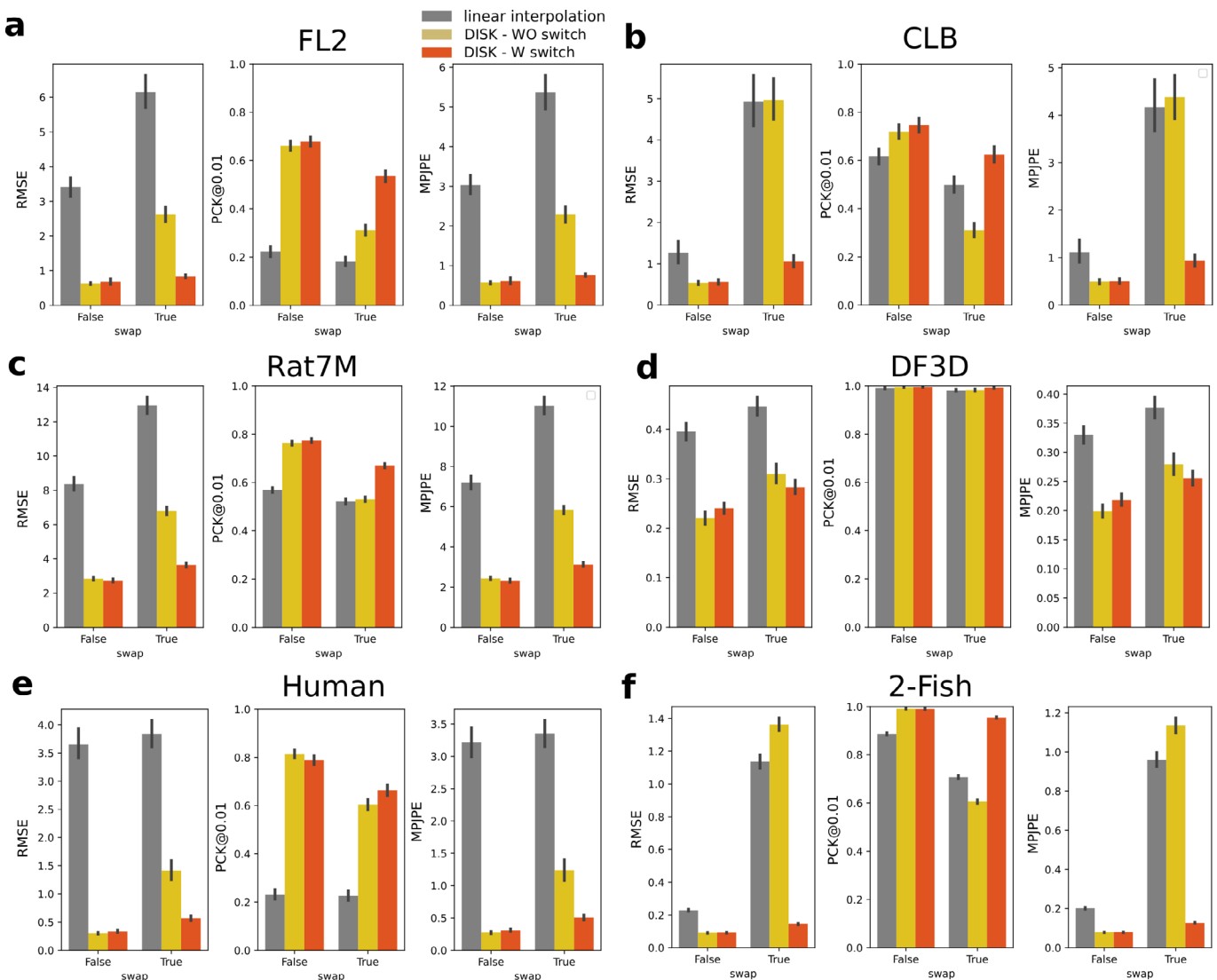

**Extended Data Fig. 6 | Additional RMSE, MPJPE and PCK@0.01 for the switch and non-switch DISK models.** CLB (**a**), 2-Mice-2D (**b**), Rat7M (**c**), DF3D (**d**), Human (**e**) and 2-Fish (**f**) datasets (original units, millimeters for all datasets except for Human dataset). The bar plots represent the mean and the error bars the 95% confidence interval.

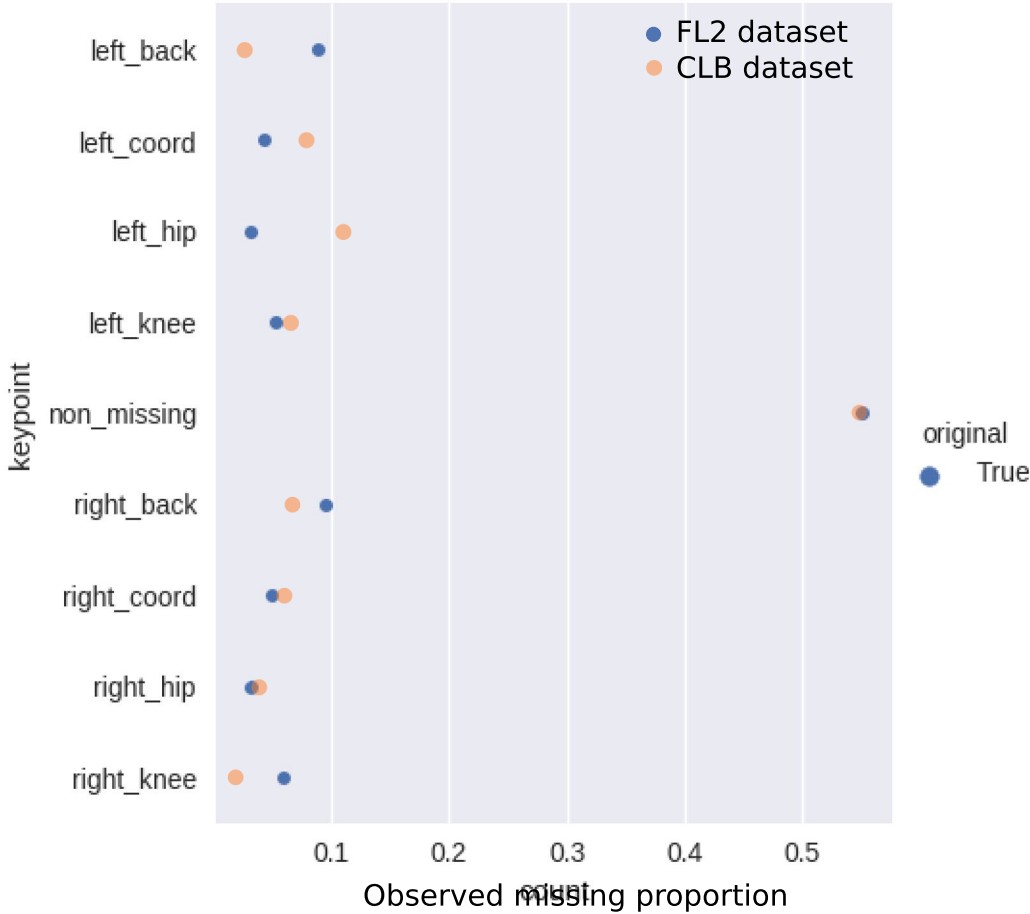

**Extended Data Fig. 7 | Observed missing proportions for each keypoint in mouse FL2 and CLB datasets.** Each keypoint is considered independently of the others to calculate its missing proportion.

**Extended Data Table 1 | Datasets**

| Dataset | N kp | Freq | Stride | Size train / val / test | Missing prop [%] |
|---|---|---|---|---|---|
| FL2 | 8 | 60 | 30 | 4,396 / 422 / 413 | 24 |
| CLB | 8 | 60 | 30 | 8,571 / 983 / 918 | 16 |
| DF3D | 38 | 100 | 5 | 2,095 / 652 / 614 | 0 |
| Human | 20 | 12 | 30 | 8,593 / 823 / 869 | 0 |
| Rat7M | 20 | 30 | 30 | 13,463 / 2,840 / 2,713 | 44 |
| 2-Fish | $2 \times 3$ | 60 | 120 | 99,029 / 13,327 / 15,705 | 6 |
| 2-Mice-2D | $2 \times 7$ | 30 | 60 | 6,820 / 986 / 622 | 0 |

*Nkp* corresponds to the number of available keypoints, *Freq* to the considered frequency of the dataset, sometimes after downsampling. The *Size train / val / test* refers to the number of 60 frames sequences (called samples) separated by a length of *Stride in the train, validation and test sets respectively*. *Missing prop* corresponds to the proportion of time points where at least one keypoint is missing.

| | |
|---|---|

# Reporting Summary

## Statistics

For all statistical analyses, confirm that the following items are present in the figure legend, table legend, main text, or Methods section.

| n/a | Confirmed | |
|---|---|---|
| ☐ | ☒ | The exact sample size (*n*) for each experimental group/condition, given as a discrete number and unit of measurement |
| ☐ | ☒ | A statement on whether measurements were taken from distinct samples or whether the same sample was measured repeatedly |
| ☐ | ☒ | The statistical test(s) used AND whether they are one- or two-sided<br>*Only common tests should be described solely by name; describe more complex techniques in the Methods section.* |
| ☐ | ☒ | A description of all covariates tested |
| ☒ | ☐ | A description of any assumptions or corrections, such as tests of normality and adjustment for multiple comparisons |
| ☐ | ☒ | A full description of the statistical parameters including central tendency (e.g. means) or other basic estimates (e.g. regression coefficient) AND variation (e.g. standard deviation) or associated estimates of uncertainty (e.g. confidence intervals) |
| ☐ | ☒ | For null hypothesis testing, the test statistic (e.g. *F*, *t*, *r*) with confidence intervals, effect sizes, degrees of freedom and *P* value noted<br>*Give P values as exact values whenever suitable.* |
| ☒ | ☐ | For Bayesian analysis, information on the choice of priors and Markov chain Monte Carlo settings |
| ☒ | ☐ | For hierarchical and complex designs, identification of the appropriate level for tests and full reporting of outcomes |
| ☐ | ☒ | Estimates of effect sizes (e.g. Cohen's *d*, Pearson's *r*), indicating how they were calculated |

*Our web collection on statistics for biologists contains articles on many of the points above.*

## Software and code

Policy information about availability of computer code

| Data collection | *Provide a description of all commercial, open source and custom code used to collect the data in this study, specifying the version used OR state that no software was used.* |
|---|---|
| Data analysis | Custom code are available at https://github.com/bozeklab/DISK (v1.0.0) and https://github.com/bozeklab/DISK_paper_code (main branch as in July 2025). DISK code base contains the method proposed in the manuscript. DISK_paper_code contains the scripts and files to reproduce the results in the paper, including running the methods DISK was compared to. Both repositories are written in Python3.9. The packages and versions needed to run the code are available in the `requirements.txt`file in each github repositories. |

For manuscripts utilizing custom algorithms or software that are central to the research but not yet described in published literature, software must be made available to editors and reviewers. We strongly encourage code deposition in a community repository (e.g. GitHub). See the Nature Portfolio guidelines for submitting code & software for further information.

## Data

Policy information about availability of data

All manuscripts must include a data availability statement. This statement should provide the following information, where applicable:

- Accession codes, unique identifiers, or web links for publicly available datasets
- A description of any restrictions on data availability
- For clinical datasets or third party data, please ensure that the statement adheres to our policy

The following original datasets used in the publication are available online: Fish (https://zenodo.org/records/10103747, doi:10.5281/zenodo.10103746), MABe (Resources section of https://www.aicrowd.com/challenges/multi-agent-behavior-representation-modeling-measurement-and-applications/problems/mabe-task-1-classical-classification), DF3D (https://dataverse.harvard.edu/dataverse/DeepFly3D), Human (CMU Mocap section at https://ericguo5513.github.io/action-to-motion//#data), Rat7M (https://figshare.com/collections/Rat 7M/5295370/3).
The mouse data (FL2 and CLB) is available upon request. Data from the same recording rig are available at https://zenodo.org/records/15493339.
Additionally to support the DISK method, we provide processed datasets and model checkpoints at https://doi.org/10.5281/zenodo.15828939.

## Human research participants

Policy information about studies involving human research participants and Sex and Gender in Research.

| | |
|---|---|
| Reporting on sex and gender | Use the terms sex (biological attribute) and gender (shaped by social and cultural circumstances) carefully in order to avoid confusing both terms. Indicate if findings apply to only one sex or gender; describe whether sex and gender were considered in study design whether sex and/or gender was determined based on self-reporting or assigned and methods used. Provide in the source data disaggregated sex and gender data where this information has been collected, and consent has been obtained for sharing of individual-level data; provide overall numbers in this Reporting Summary. Please state if this information has not been collected. Report sex- and gender-based analyses where performed, justify reasons for lack of sex- and gender-based analysis. |
| Population characteristics | Describe the covariate-relevant population characteristics of the human research participants (e.g. age, genotypic information, past and current diagnosis and treatment categories). If you filled out the behavioural & social sciences study design questions and have nothing to add here, write "See above." |
| Recruitment | Describe how participants were recruited. Outline any potential self-selection bias or other biases that may be present and how these are likely to impact results. |
| Ethics oversight | Identify the organization(s) that approved the study protocol. |

Note that full information on the approval of the study protocol must also be provided in the manuscript.

# Field-specific reporting

Please select the one below that is the best fit for your research. If you are not sure, read the appropriate sections before making your selection.

☒ Life sciences        ☐ Behavioural & social sciences        ☐ Ecological, evolutionary & environmental sciences

For a reference copy of the document with all sections, see nature.com/documents/nr-reporting-summary-flat.pdf

# Life sciences study design

All studies must disclose on these points even when the disclosure is negative.

| | |
|---|---|
| Sample size | Sample sizes are available in Ext. Table 1 and in specific figure legends. |
| Data exclusions | For benchmarking neural architectures and other methods on the Mouse FL2 and CLB datasets, the ankles were excluded because they were missing too frequently. |
| Replication | Given the data and provided code, results can be replicated. The fake gaps are randomly generated at each run of the train or test scripts. So the number and nature of gaps used in the testing phase is not deterministic. We made sure this had only a small impact on the performance of the models (cf Fig. 1a and Ext. Fig. 1 a and b). |
| Randomization | The allocation of samples to the train / validation / test splits was done randomly ona file basis, except for Rat7M dataset that were containing only 6 long recordings (cf Methods section). In particular the splits were not chosen to optimize performance. |
| Blinding | We did not collect data ourselves. |

# Behavioural & social sciences study design

All studies must disclose on these points even when the disclosure is negative.

| | |
|---|---|
| Study description | *Briefly describe the study type including whether data are quantitative, qualitative, or mixed-methods (e.g. qualitative cross-sectional, quantitative experimental, mixed-methods case study).* |
| Research sample | *State the research sample (e.g. Harvard university undergraduates, villagers in rural India) and provide relevant demographic information (e.g. age, sex) and indicate whether the sample is representative. Provide a rationale for the study sample chosen. For studies involving existing datasets, please describe the dataset and source.* |
| Sampling strategy | *Describe the sampling procedure (e.g. random, snowball, stratified, convenience). Describe the statistical methods that were used to predetermine sample size OR if no sample-size calculation was performed, describe how sample sizes were chosen and provide a rationale for why these sample sizes are sufficient. For qualitative data, please indicate whether data saturation was considered, and what criteria were used to decide that no further sampling was needed.* |
| Data collection | *Provide details about the data collection procedure, including the instruments or devices used to record the data (e.g. pen and paper, computer, eye tracker, video or audio equipment) whether anyone was present besides the participant(s) and the researcher, and whether the researcher was blind to experimental condition and/or the study hypothesis during data collection.* |
| Timing | *Indicate the start and stop dates of data collection. If there is a gap between collection periods, state the dates for each sample cohort.* |
| Data exclusions | *If no data were excluded from the analyses, state so OR if data were excluded, provide the exact number of exclusions and the rationale behind them, indicating whether exclusion criteria were pre-established.* |
| Non-participation | *State how many participants dropped out/declined participation and the reason(s) given OR provide response rate OR state that no participants dropped out/declined participation.* |
| Randomization | *If participants were not allocated into experimental groups, state so OR describe how participants were allocated to groups, and if allocation was not random, describe how covariates were controlled.* |

# Ecological, evolutionary & environmental sciences study design

All studies must disclose on these points even when the disclosure is negative.

| | |
|---|---|
| Study description | *Briefly describe the study. For quantitative data include treatment factors and interactions, design structure (e.g. factorial, nested, hierarchical), nature and number of experimental units and replicates.* |
| Research sample | *Describe the research sample (e.g. a group of tagged Passer domesticus, all Stenocereus thurberi within Organ Pipe Cactus National Monument), and provide a rationale for the sample choice. When relevant, describe the organism taxa, source, sex, age range and any manipulations. State what population the sample is meant to represent when applicable. For studies involving existing datasets, describe the data and its source.* |
| Sampling strategy | *Note the sampling procedure. Describe the statistical methods that were used to predetermine sample size OR if no sample-size calculation was performed, describe how sample sizes were chosen and provide a rationale for why these sample sizes are sufficient.* |
| Data collection | *Describe the data collection procedure, including who recorded the data and how.* |
| Timing and spatial scale | *Indicate the start and stop dates of data collection, noting the frequency and periodicity of sampling and providing a rationale for these choices. If there is a gap between collection periods, state the dates for each sample cohort. Specify the spatial scale from which the data are taken* |
| Data exclusions | *If no data were excluded from the analyses, state so OR if data were excluded, describe the exclusions and the rationale behind them, indicating whether exclusion criteria were pre-established.* |
| Reproducibility | *Describe the measures taken to verify the reproducibility of experimental findings. For each experiment, note whether any attempts to repeat the experiment failed OR state that all attempts to repeat the experiment were successful.* |
| Randomization | *Describe how samples/organisms/participants were allocated into groups. If allocation was not random, describe how covariates were controlled. If this is not relevant to your study, explain why.* |
| Blinding | *Describe the extent of blinding used during data acquisition and analysis. If blinding was not possible, describe why OR explain why blinding was not relevant to your study.* |

Did the study involve field work?  ☐ Yes  ☐ No

# Field work, collection and transport

| | |
|---|---|
| Field conditions | *Describe the study conditions for field work, providing relevant parameters (e.g. temperature, rainfall).* |
| Location | *State the location of the sampling or experiment, providing relevant parameters (e.g. latitude and longitude, elevation, water depth).* |
| Access & import/export | *Describe the efforts you have made to access habitats and to collect and import/export your samples in a responsible manner and in compliance with local, national and international laws, noting any permits that were obtained (give the name of the issuing authority, the date of issue, and any identifying information).* |
| Disturbance | *Describe any disturbance caused by the study and how it was minimized.* |

# Reporting for specific materials, systems and methods

We require information from authors about some types of materials, experimental systems and methods used in many studies. Here, indicate whether each material, system or method listed is relevant to your study. If you are not sure if a list item applies to your research, read the appropriate section before selecting a response.

## Materials & experimental systems

| n/a | Involved in the study |
|---|---|
| ☒ | ☐ Antibodies |
| ☒ | ☐ Eukaryotic cell lines |
| ☒ | ☐ Palaeontology and archaeology |
| ☒ | ☐ Animals and other organisms |
| ☒ | ☐ Clinical data |
| ☒ | ☐ Dual use research of concern |

## Methods

| n/a | Involved in the study |
|---|---|
| ☒ | ☐ ChIP-seq |
| ☒ | ☐ Flow cytometry |
| ☒ | ☐ MRI-based neuroimaging |

## Antibodies

| | |
|---|---|
| Antibodies used | *Describe all antibodies used in the study; as applicable, provide supplier name, catalog number, clone name, and lot number.* |
| Validation | *Describe the validation of each primary antibody for the species and application, noting any validation statements on the manufacturer's website, relevant citations, antibody profiles in online databases, or data provided in the manuscript.* |

## Eukaryotic cell lines

Policy information about cell lines and Sex and Gender in Research

| | |
|---|---|
| Cell line source(s) | *State the source of each cell line used and the sex of all primary cell lines and cells derived from human participants or vertebrate models.* |
| Authentication | *Describe the authentication procedures for each cell line used OR declare that none of the cell lines used were authenticated.* |
| Mycoplasma contamination | *Confirm that all cell lines tested negative for mycoplasma contamination OR describe the results of the testing for mycoplasma contamination OR declare that the cell lines were not tested for mycoplasma contamination.* |
| Commonly misidentified lines<br>(See ICLAC register) | *Name any commonly misidentified cell lines used in the study and provide a rationale for their use.* |

## Palaeontology and Archaeology

| | |
|---|---|
| Specimen provenance | *Provide provenance information for specimens and describe permits that were obtained for the work (including the name of the issuing authority, the date of issue, and any identifying information). Permits should encompass collection and, where applicable, export.* |
| Specimen deposition | *Indicate where the specimens have been deposited to permit free access by other researchers.* |

| Dating methods | *If new dates are provided, describe how they were obtained (e.g. collection, storage, sample pretreatment and measurement), where they were obtained (i.e. lab name), the calibration program and the protocol for quality assurance OR state that no new dates are provided.* |

☐ Tick this box to confirm that the raw and calibrated dates are available in the paper or in Supplementary Information.

| Ethics oversight | *Identify the organization(s) that approved or provided guidance on the study protocol, OR state that no ethical approval or guidance was required and explain why not.* |

Note that full information on the approval of the study protocol must also be provided in the manuscript.

# Animals and other research organisms

Policy information about studies involving animals; ARRIVE guidelines recommended for reporting animal research, and Sex and Gender in Research

| Laboratory animals | *For laboratory animals, report species, strain and age OR state that the study did not involve laboratory animals.* |

| Wild animals | *Provide details on animals observed in or captured in the field; report species and age where possible. Describe how animals were caught and transported and what happened to captive animals after the study (if killed, explain why and describe method; if released, say where and when) OR state that the study did not involve wild animals.* |

| Reporting on sex | *Indicate if findings apply to only one sex; describe whether sex was considered in study design, methods used for assigning sex. Provide data disaggregated for sex where this information has been collected in the source data as appropriate; provide overall numbers in this Reporting Summary. Please state if this information has not been collected. Report sex-based analyses where performed, justify reasons for lack of sex-based analysis.* |

| Field-collected samples | *For laboratory work with field-collected samples, describe all relevant parameters such as housing, maintenance, temperature, photoperiod and end-of-experiment protocol OR state that the study did not involve samples collected from the field.* |

| Ethics oversight | *Identify the organization(s) that approved or provided guidance on the study protocol, OR state that no ethical approval or guidance was required and explain why not.* |

Note that full information on the approval of the study protocol must also be provided in the manuscript.

# Clinical data

Policy information about clinical studies

All manuscripts should comply with the ICMJE guidelines for publication of clinical research and a completed CONSORT checklist must be included with all submissions.

| Clinical trial registration | *Provide the trial registration number from ClinicalTrials.gov or an equivalent agency.* |

| Study protocol | *Note where the full trial protocol can be accessed OR if not available, explain why.* |

| Data collection | *Describe the settings and locales of data collection, noting the time periods of recruitment and data collection.* |

| Outcomes | *Describe how you pre-defined primary and secondary outcome measures and how you assessed these measures.* |

# Dual use research of concern

Policy information about dual use research of concern

## Hazards

Could the accidental, deliberate or reckless misuse of agents or technologies generated in the work, or the application of information presented in the manuscript, pose a threat to:

No | Yes
☒ ☐ Public health
☒ ☐ National security
☒ ☐ Crops and/or livestock
☒ ☐ Ecosystems
☒ ☐ Any other significant area

## Experiments of concern

Does the work involve any of these experiments of concern:

No | Yes
☒ | ☐ Demonstrate how to render a vaccine ineffective
☒ | ☐ Confer resistance to therapeutically useful antibiotics or antiviral agents
☒ | ☐ Enhance the virulence of a pathogen or render a nonpathogen virulent
☒ | ☐ Increase transmissibility of a pathogen
☒ | ☐ Alter the host range of a pathogen
☒ | ☐ Enable evasion of diagnostic/detection modalities
☒ | ☐ Enable the weaponization of a biological agent or toxin
☒ | ☐ Any other potentially harmful combination of experiments and agents

# ChIP-seq

## Data deposition

☐ Confirm that both raw and final processed data have been deposited in a public database such as GEO.

☐ Confirm that you have deposited or provided access to graph files (e.g. BED files) for the called peaks.

Data access links
*May remain private before publication.*

*For "Initial submission" or "Revised version" documents, provide reviewer access links. For your "Final submission" document, provide a link to the deposited data.*

Files in database submission

*Provide a list of all files available in the database submission.*

Genome browser session
(e.g. UCSC)

*Provide a link to an anonymized genome browser session for "Initial submission" and "Revised version" documents only, to enable peer review. Write "no longer applicable" for "Final submission" documents.*

## Methodology

Replicates

*Describe the experimental replicates, specifying number, type and replicate agreement.*

Sequencing depth

*Describe the sequencing depth for each experiment, providing the total number of reads, uniquely mapped reads, length of reads and whether they were paired- or single-end.*

Antibodies

*Describe the antibodies used for the ChIP-seq experiments; as applicable, provide supplier name, catalog number, clone name, and lot number.*

Peak calling parameters

*Specify the command line program and parameters used for read mapping and peak calling, including the ChIP, control and index files used.*

Data quality

*Describe the methods used to ensure data quality in full detail, including how many peaks are at FDR 5% and above 5-fold enrichment.*

Software

*Describe the software used to collect and analyze the ChIP-seq data. For custom code that has been deposited into a community repository, provide accession details.*

# Flow Cytometry

## Plots

Confirm that:

☐ The axis labels state the marker and fluorochrome used (e.g. CD4-FITC).

☐ The axis scales are clearly visible. Include numbers along axes only for bottom left plot of group (a 'group' is an analysis of identical markers).

☐ All plots are contour plots with outliers or pseudocolor plots.

☐ A numerical value for number of cells or percentage (with statistics) is provided.

## Methodology

Sample preparation

*Describe the sample preparation, detailing the biological source of the cells and any tissue processing steps used.*

Instrument

*Identify the instrument used for data collection, specifying make and model number.*

| Software | *Describe the software used to collect and analyze the flow cytometry data. For custom code that has been deposited into a community repository, provide accession details.* |
|---|---|
| Cell population abundance | *Describe the abundance of the relevant cell populations within post-sort fractions, providing details on the purity of the samples and how it was determined.* |
| Gating strategy | *Describe the gating strategy used for all relevant experiments, specifying the preliminary FSC/SSC gates of the starting cell population, indicating where boundaries between "positive" and "negative" staining cell populations are defined.* |

☐ Tick this box to confirm that a figure exemplifying the gating strategy is provided in the Supplementary Information.

# Magnetic resonance imaging

## Experimental design

| Design type | *Indicate task or resting state; event-related or block design.* |
|---|---|
| Design specifications | *Specify the number of blocks, trials or experimental units per session and/or subject, and specify the length of each trial or block (if trials are blocked) and interval between trials.* |
| Behavioral performance measures | *State number and/or type of variables recorded (e.g. correct button press, response time) and what statistics were used to establish that the subjects were performing the task as expected (e.g. mean, range, and/or standard deviation across subjects).* |

## Acquisition

| Imaging type(s) | *Specify: functional, structural, diffusion, perfusion.* |
|---|---|
| Field strength | *Specify in Tesla* |
| Sequence & imaging parameters | *Specify the pulse sequence type (gradient echo, spin echo, etc.), imaging type (EPI, spiral, etc.), field of view, matrix size, slice thickness, orientation and TE/TR/flip angle.* |
| Area of acquisition | *State whether a whole brain scan was used OR define the area of acquisition, describing how the region was determined.* |

Diffusion MRI    ☐ Used    ☐ Not used

## Preprocessing

| Preprocessing software | *Provide detail on software version and revision number and on specific parameters (model/functions, brain extraction, segmentation, smoothing kernel size, etc.).* |
|---|---|
| Normalization | *If data were normalized/standardized, describe the approach(es): specify linear or non-linear and define image types used for transformation OR indicate that data were not normalized and explain rationale for lack of normalization.* |
| Normalization template | *Describe the template used for normalization/transformation, specifying subject space or group standardized space (e.g. original Talairach, MNI305, ICBM152) OR indicate that the data were not normalized.* |
| Noise and artifact removal | *Describe your procedure(s) for artifact and structured noise removal, specifying motion parameters, tissue signals and physiological signals (heart rate, respiration).* |
| Volume censoring | *Define your software and/or method and criteria for volume censoring, and state the extent of such censoring.* |

## Statistical modeling & inference

| Model type and settings | *Specify type (mass univariate, multivariate, RSA, predictive, etc.) and describe essential details of the model at the first and second levels (e.g. fixed, random or mixed effects; drift or auto-correlation).* |
|---|---|
| Effect(s) tested | *Define precise effect in terms of the task or stimulus conditions instead of psychological concepts and indicate whether ANOVA or factorial designs were used.* |

Specify type of analysis:    ☐ Whole brain    ☐ ROI-based    ☐ Both

| Statistic type for inference<br>(See Eklund et al. 2016) | *Specify voxel-wise or cluster-wise and report all relevant parameters for cluster-wise methods.* |
|---|---|
| Correction | *Describe the type of correction and how it is obtained for multiple comparisons (e.g. FWE, FDR, permutation or Monte Carlo).* |

## Models & analysis

**Functional and/or effective connectivity**

*Report the measures of dependence used and the model details (e.g. Pearson correlation, partial correlation, mutual information).*

**Graph analysis**

*Report the dependent variable and connectivity measure, specifying weighted graph or binarized graph, subject- or group-level, and the global and/or node summaries used (e.g. clustering coefficient, efficiency, etc.).*

**Multivariate modeling and predictive analysis**

*Specify independent variables, features extraction and dimension reduction, model, training and evaluation metrics.*

