## [Peer Review File · Nature Methods]

Deep Imputation for Skeleton Data (DISK) for Behavioral Science

Corresponding Author: Dr France Rose

Version 0:

Decision Letter:

10th Jul 2024

Dear Dr Rose,

Thank you for your patience. Your Article, "Deep Imputation for Skeleton Data (DISK) for Behavioral Science", has now been seen by two reviewers. As you will see from their comments below, although the reviewers find your work of considerable potential interest, they have raised a number of concerns. We are interested in the possibility of publishing your paper in Nature Methods, but would like to consider your response to these concerns before we reach a final decision on publication.

We therefore invite you to revise your manuscript to address these concerns. As you will see one of the reviewers mentioned some human studies. As Nature Methods is focused on methods for basic research, please do make sure that the focus of your manuscript remains on animals.

Link Redacted

We hope to receive your revised paper within 2-3 months. If you cannot send it within this time, please let us know. In this event, we will still be happy to reconsider your paper at a later date so long as nothing similar has been accepted for publication at Nature Methods or published elsewhere.

OPEN SCIENCE REQUIREMENTS

REPORTING SUMMARY AND EDITORIAL POLICY CHECKLISTS

DATA AVAILABILITY

All novel DNA and RNA sequencing data, protein sequences, genetic polymorphisms, linked genotype and phenotype data, gene expression data, macromolecular structures, and proteomics data must be deposited in a publicly accessible database, and accession codes and associated hyperlinks must be provided in the "Data Availability" section.

CODE AVAILABILITY

Please include a "Code Availability" subsection in the Online Methods which details how your custom code is made available. Only in rare cases (where code is not central to the main conclusions of the paper) is the statement "available upon request" allowed (and reasons should be specified).

MATERIALS AVAILABILITY

ORCID

Best regards,
Nina

Nina Vogt, PhD
Senior Editor
Nature Methods

Reviewers' Comments:

Reviewer #1 (Remarks to the Author):

A. The paper discusses a novel method for animal and human pose estimation. The method learns spatio-temporal relations among the keypoints of the subject's skeleton by augmenting the training dataset by randomly masking a subset of keypoints and thus emulating occlusion in real-world data. The authors conducted experiments on diverse animal subjects (humans, mice, rats, flies, and fish). Furthermore, they analyzed the latent embedding of their model to classify behaviors.

B. The proposed method is novel and potentially significant (significance could be shown with an additional experiment). The following claims for novelty should be reworded:

"no general missing data imputation method has been developed for skeleton data" Line 88

"We demonstrate the first to our knowledge deep learning approach to impute missing coordinates in 2D and 3D skeleton data." Line 551

Prior works exist to learn spatio-temporal relations among skeletal data in an unsupervised manner [3,4]. Randomly masking of keypoints to emulate real-world occlusion was explored by Patel et al., 2023 [4].

"Two of them, GRU and TCN were designed for time-series analysis, but have not been tested specifically on skeleton data." Line 321

Gholami et al. [1] use GRU, and Pavllo et al. [2] use TCN for skeletal data.

C.

Strengths of Methodology:

- The authors explored several different neural network architectures: Temporal CNNs, GRU, GCNs, and Transformers.
- They propose a framework that predicts the estimated error in missing skeletal data, which can further be used for thresholding and filtering.
- The authors use a latent space embedding paired with kinematic information to classify behaviors.

Weaknesses in Experimental Design:

- The paper lacks a quantitative comparison with prior works on benchmark datasets. We recommend that the authors take off-the-shelf pose estimation methods with easily available model checkpoints, such as provided by references [5], [6], and [7], and evaluate whether their method improves the accuracy using the standard pose estimation metrics.
- The paper does not use standard pose estimation metrics, such as PCK, MPJPE, and P-MPJPE [8,9].

D. See F about additional experiments and metrics.

E. The use of the metrics mentioned above would better represent the reconstruction quality of the proposed method and show whether its results are competitive with existing techniques and thus provides an improvement that warrants publication in Nature Methods.

F. The quantitative comparison with prior works on benchmark datasets that we recommended above should be relatively easy to

do and not require a major revision of the paper. It would result in a few additional paragraphs and new result table.

The code needs to be fixed so that it can be run as suggested:

The drive link in the reporting summary does not work.

The `impute_dataset.py` file for the Rat7M dataset with the pre-trained checkpoint did not work.

Some crucial keys in `“hydra/config.yaml”` are missing.

G. References mentioned in this review:

1 Gholami, Mohsen, et al. “Self-Supervised 3D Human Pose Estimation from Video.” *Neurocomputing*, Mar. 2022, <https://doi.org/10.1016/j.neucom.2022.02.076>. Accessed 6 Mar. 2022.

2 Pavlo, Dario, et al. “3D Human Pose Estimation in Video with Temporal Convolutions and Semi-Supervised Training.” *ArXiv.org*, 29 Mar. 2019, arxiv.org/abs/1811.11742.

3 Ruiz, Alejandro Hernandez, et al. “Human Motion Prediction via Spatio-Temporal Inpainting.” *ArXiv.org*, 28 Oct. 2019, arxiv.org/abs/1812.05478. Accessed 30 May 2024.

4 Patel, M., Gu, Y., Carstensen, L.C. et al. Animal Pose Tracking: 3D Multimodal Dataset and Token-based Pose Optimization. *Int J Comput Vis* 131, 514–530 (2023). <https://doi.org/10.1007/s11263-022-01714-5>

5 Iskakov, K., Burkov, E., Lempitsky, V., & Malkov, Y. (2019). Learnable Triangulation of Human Pose. https://openaccess.thecvf.com/content_ICCV_2019/html/Iskakov_Learnable_Triangulation_of_Human_Pose_ICCV_2019_paper.html

6 He, Y., Yan, R., Fragkiadaki, K., & Yu, S. I. (2020). Epipolar Transformers. https://openaccess.thecvf.com/content_CVPR_2020/html/He_Epipolar_Transformers_CVPR_2020_paper.html

7 Mathis, A., Mamidanna, P., Cury, K.M. et al. DeepLabCut: markerless pose estimation of user-defined body parts with deep learning. *Nat Neurosci* 21, 1281–1289 (2018). <https://doi.org/10.1038/s41593-018-0209-y>

8 Mehta, D., Rhodin, H., Casas, D., Fua, P., Sotnychenko, O., Xu, W., & Theobalt, C. (2017). Monocular 3D Human Pose Estimation In The Wild Using Improved CNN Supervision. <https://vcai.mpi-inf.mpg.de/3dhp-dataset/>

9 MODEC: Multimodal Decomposable Models for Human Pose Estimation. (2013, June 1). *IEEE Conference Publication | IEEE Xplore*. <https://ieeexplore.ieee.org/document/6619315>

H. Clarity and context are good.

Reviewer #1 (Remarks on code availability):

The drive link in the reporting summary does not work.

The `impute_dataset.py` file for the Rat7M dataset with the pre-trained checkpoint did not work.

Some crucial keys in `“hydra/config.yaml”` are missing.

Reviewer #2 (Remarks to the Author):

This study by Rose et al aims to address missing data in kinematic tracking and pose estimation methods and offers a method for imputing 2D and 3D tracking data as well as assessing the accuracy of imputation in order to prepare data for downstream analyses. There is a growing need for methods to accurately impute missing and spurious skeleton tracking data due to the growing use of markerless tracking methods in animal behavior studies, and a simple tool that could be run on any kinematic data to clean up missing keypoints would be of great use. However, I have several reservations about the method and claims presented here.

Major Issues:

While the claim here is that the keypoint dependencies and dynamics are used to impute missing tracking data, why are the snippets for training limited to ~60 frames? Does this differ if imaging at a higher frame rate? What happens when dropouts are longer than this but could easily be restored with other methods (even simple interpolation), for example when an animal is still but one marker or keypoint is not visible.

One major issue with tracking data (especially in marker-based tracking) is the tendency to switch keypoints (even more so in social tracking), have the authors done anything to address how to identify and correct these flips?

The strongly worded claim that ‘no general missing data imputation method has been developed for skeleton data’ sends one looking for counterexamples - would something like the method developed here for one of the manuscript’s use cases (<https://github.com/diegoaldarondo/MarkerBasedImputation>) not qualify as it is not general enough? In that case could a direct comparison be worthwhile to show how this method performs against a method designed with a particular dataset in mind?

Accuracy metrics for interpolation will always be biased to common postures that might be more easily tracked - I would like to see how the algorithm performs on difficult/ less common postures that might actually limit downstream analyses. Given that some of these datasets are annotated for behavior - has there been any attempt to test this method on a subset of behaviors during which tracking fails more commonly? Introducing missing data at random might bias the results to appearing more accurate than they really are for given tricky subsets of behavior.

There is a tradeoff between recovering missing data via interpolation and training better models that incorporate skeletal information (especially for 3D data with good calibration) - a discussion on this, limitations of this method, and future directions that could incorporate this type interpolation with tracking seems warranted.

Minor Issues:

115 - A keypoint tracking network can be forced to have no missing values - and can always be retrained to be more accurate - so the fear of losing data and needing to take new recordings when the videos exist seems unfounded.

Figure 1 - Not sure why both panels a and c exist here, also for the Rat7M dataset - does this refer to the total fraction of frames with missing keypoints or the total fraction of missing keypoints? Since the authors performed their own interpolation, how does the method here compare?

Figure 2 - How is RMSE calculated for 2D data? The individual panels here seem unnecessary - they could be combined into a stacked plot and include a much longer time window to give a sense of how the interpolation actually looks over time. In b-i, what is the ground truth for any of the tracks (and does b really need 7 examples of tracks with no missing data and no interpolation)?

Figures 5 and 6 seem like an incomplete departure to a different story altogether - it seems obvious that latent spaces encoding posture and dynamics would segregate based on type of behavior being performed, is the point here that the latents encode something about posture and dynamics? What's going on with the 'walk' skeleton in Figure 5a? Either way, it seems like a distraction from the main point.

293 - the language here suggests that a larger number of steps detected is good - but why not use an artificial dataset that has a true 'correct' number of steps as well as dynamics and then perform the test on that? If this was done in 3D then a true distance to the target could be calculated at every point in time.

Version 1:

Decision Letter:

22nd Jan 2025

Dear Dr Rose,

Thank you for your letter detailing how you would respond to the reviewer concerns regarding your Article, "Deep Imputation for Skeleton Data (DISK) for Behavioral Science". We have decided to invite you to revise your manuscript as you have outlined, before we reach a final decision on publication.

Link Redacted

We hope to receive your revised paper within 1-2 months. If you cannot send it within this time, please let us know. In this event, we will still be happy to reconsider your paper at a later date so long as nothing similar has been accepted for publication at Nature Methods or published elsewhere.

OPEN SCIENCE REQUIREMENTS

REPORTING SUMMARY AND EDITORIAL POLICY CHECKLISTS

EXTENDED DATA FIGURES

DATA AVAILABILITY

CODE AVAILABILITY

Please include a "Code Availability" subsection in the Online Methods which details how your custom code is made available. Only in rare cases (where code is not central to the main conclusions of the paper) is the statement "available upon request" allowed (and reasons should be specified).

MATERIALS AVAILABILITY

ORCID

Best regards,
Nina

Nina Vogt, PhD
Senior Editor
Nature Methods

Reviewers' Comments:

Reviewer #1 (Remarks to the Author):

The paper discusses a method for learning spatio-temporal relations among the keypoints of the skeleton by augmenting the training dataset by randomly masking a subset of keypoints and thus emulating occlusion in real-world data. The authors conducted experiments on diverse animal subjects (humans, mice, rats, flies, and fish). Furthermore, they analyze the latent embedding of their model to classify behaviors.

- The authors explored several different neural network architectures: Temporal CNNs, GRU, GCNs, and Transformers.
- They propose a framework that predicts the estimated error in missing skeletal data, which can further be used for thresholding and filtering.
- The authors use a latent space embedding paired with kinematic information to classify behaviors.
- The paper uses well-defined standard pose estimation metrics.
- The authors demonstrate their method on Mouse FL2 dataset by predicting missing data; which made their results closer to manual counts performed by experts.

The code issue has been fixed. The provided notebook is easy to follow and covers all relevant use cases. Training a model and running it in inference mode seem straightforward. The code is reasonably well-documented and easy to follow. The authors have also provided an FAQ that covers what the hyperparameters do and how to change them.

One remaining issue:

Line 92: There is however no general method for imputation of missing keypoints in skeleton motion data that would work on any type of animal skeleton, including one or more individuals, and be independent of the keypoint tracking method.

Both OptiPose and Keypoint-MoSeq would work for any type of animal skeleton. OptiPose can be run individually for each instance of animal independently (however lacks data association reasoning).

In summary, the authors addressed all of our concerns and added relevant metrics and comparisons with prior works.

Reviewer #2 (Remarks to the Author):

The edited manuscript by Rose et al addressed some of my comments and overall presents a method that may be useful in imputing tracking data from behavioral studies. The main issues previously I raised about the proposed method were longer

dropouts and missing data, keypoint switches, and a lack of acknowledgement for previously published methods. DISK is suitable for many types of skeletons, including multi-animal set-ups, and the authors have demonstrated improvement in tracking using DISK in several use cases. For this revision, the authors have added comparison to other methods, more accuracy metrics, and some additional experiments testing keypoint switching. The comparison to Kp-Moseq and Optipose, which I had not originally considered, raised more questions about why the others methods have such poor outputs outside of interpolation gaps. The test of switch keypoints shows that accuracy is improved and that the uncertainty of the model might be useful in a more comprehensive switching detector and interpolator. However, as with most of the quantification in the original manuscript and the rebuttal, I would like to see summary statistics of switch types (within single animal, multi-animal, across species) across many lengths and types of switches instead of just a few traces.

I appreciate the authors have produced a much more generalizable method, but the code base github.com/diegolaldarondo/MarkerBasedImputation was published as part of the CAPTURE paper, from which this manuscript uses data (Rat7M)! Arguing that it was never published or used in published results is misguided, unless there is something I am missing here. The authors also argue that the Rat7M data was not further interpolated in the rebuttal - I think this is directly addressed in the CAPTURE paper and supplement.

I am also still very confused by the argument that 'validation on real data is more relevant compared to artificially generated step trajectories' as an argument against generating artificial, corrupting it, and then testing DISK on different types of dynamics and fraction of missing data. This would allow for a broader understanding of the limits of the types of movements DISK is good at capturing and interpolating, instead of limiting imputation results to a single type of behavioral difference in a very particular test case. In the same vein, while the result that the multi-animal interpolation in the fish data is better when they are close - this is a result that contradicts most multi-animal tracking wisdom and could also be properly explored by generating fake multi-animal trajectories and testing the method on balanced sets of deleted data.

Overall, the original manuscript and rebuttal have convinced me that DISK could be an incremental advance in imputation, but that the comparison to and scope within the broader context of marker-based and marker-less tracking is limited without additional demonstration of ease of use and accuracy.

Version 2:

Decision Letter:

Our ref: NMETH-A56101B

16th May 2025

Dear Dr. Rose,

Thank you for submitting your revised manuscript "Deep Imputation for Skeleton Data (DISK) for Behavioral Science" (NMETH-A56101B). It has now been seen by the original referees and their comments are below. The reviewers find that the paper has improved in revision, and therefore we'll be happy in principle to publish it in Nature Methods, pending minor revisions to satisfy the referees' final requests and to comply with our editorial and formatting guidelines.

TRANSPARENT PEER REVIEW

ORCID

Best regards,

Nina

Nina Vogt, PhD
Senior Editor
Nature Methods

Reviewer #1 (Remarks to the Author):

A.-H. See previous review.

I read the comments and questions of the other reviewer. The authors' responses are clear and thoughtful. The additional material that was added to the paper due to the review process is helpful.

Some wording & spelling suggestions:

Abstract: "We found that transformer outperforms other architectures..."

When using "transformer" as the name of a specific transformer architecture, it should be capitalized: "Transformer." If the authors mean a generic transformer architecture, an article is needed: "We found that a transformer outperforms other architectures..."

OptiPose should be consistently spelled with a capital P. The current document uses both OptiPose and Optipose.

"At test time, to compare the performance of on exactly the same sequences," ->

At test time, to compare the performance of the models on exactly the same sequences,

Reviewer #1 (Remarks on code availability):

As mentioned in the initial review, the provided notebook is easy to follow and covers all relevant use cases. Training a model and running it in inference mode seem straightforward. The code is reasonably well-documented and easy to follow. The authors have also provided an FAQ that covers what the hyperparameters do and how to change them.

Reviewer #2 (Remarks to the Author):

This edited manuscript addressed all of the questions I raised during previous reviews. I appreciate the thorough rebuttal and effort in implementing the MBI comparison and in all of the additional analyses that addressed my questions about generated datasets, keypoint switching, and in general clarification on the implementation of DISK and how it differs and is a complimentary tool to pose tracking. I also appreciate the well-commented code repositories.

Version 3:

Decision Letter:

1st Oct 2025

Dear France,

I am pleased to inform you that your Article, "Deep Imputation for Skeleton Data (DISK) for Behavioral Science", has now been accepted for publication in Nature Methods. The received and accepted dates will be May 2nd, 2024 and October 1st, 2025. This note is intended to let you know what to expect from us over the next month or so, and to let you know where to address any further questions.

Over the next few weeks, your paper will be copyedited to ensure that it conforms to Nature Methods style. Once your paper is typeset, you will receive an email with a link to choose the appropriate publishing options for your paper and our Author Services team will be in touch regarding any additional information that may be required. It is extremely important that you let us know now whether you will be difficult to contact over the next month. If this is the case, we ask that you send us the contact information (email, phone and fax) of someone who will be able to check the proofs and deal with any last-minute problems.

Authors may need to take specific actions to achieve compliance with funder and institutional open access mandates. If your research is supported by a funder that requires immediate open access (e.g. according to [Plan S principles](https://www.springernature.com/gp/open-science/plan-s-compliance) or the [NIH public access policy](https://www.springernature.com/gp/open-science/us-federal-agency-compliance)) then you should select the gold OA route, and we will direct you to the compliant route where possible. Because authors warrant under our

subscription licensing terms that they haven't committed to licensing any version of their article under a licence inconsistent with the terms of our agreement – including the applicable embargo period – publication under the subscription model isn't suitable for authors whose funders require no embargo.

If you are active on Twitter/X or Bluesky, please e-mail me your and your coauthors' handles so that we may tag you when the paper is published.

Best regards,
Nina

Nina Vogt, PhD
Senior Editor
Nature Methods

** Visit the Springer Nature Editorial and Publishing website at http://editorial-jobs.springernature.com?utm_source=ejp_NMeth_email&utm_medium=ejp_NMeth_email&utm_campaign=ejp_Nmeth for more information about our career opportunities. If you have any questions please click [here](mailto:editorial.publishing.jobs@springernature.com).

Dear Nina Vogt, dear Reviewers,

We thank the Reviewers' for insightful comments that helped us to improve the quality of our manuscript. In the following pages you will find a point-by-point response to the comments. Additionally, revisions in the manuscript are marked in red font. In the revision we focused in particular on the three major points raised by the Reviewers:

1. expanding the discussion of other related works and adding a comparison with the published methods *OptiPose* and *Keypoint-MoSeq*,
2. testing robustness of DISK against keypoint switching,
3. implementing two more accuracy metrics.

We hope the revised manuscript is now suitable for publication in Nature Methods. We look forward to hearing from you.

Sincerely,

France ROSE and Katarzyna BOZEK on behalf of the authors.

Reviewer #1 (Remarks to the Author):

A. The paper discusses a novel method for animal and human pose estimation. The method learns spatio-temporal relations among the keypoints of the subject’s skeleton by augmenting the training dataset by randomly masking a subset of keypoints and thus emulating occlusion in real-world data. The authors conducted experiments on diverse animal subjects (humans, mice, rats, flies, and fish). Furthermore, they analyzed the latent embedding of their model to classify behaviors.

B. The proposed method is novel and potentially significant (significance could be shown with an additional experiment). The following claims for novelty should be reworded:

- "no general missing data imputation method has been developed for skeleton data" Line 88
- "We demonstrate the first to our knowledge deep learning approach to impute missing coordinates in 2D and 3D skeleton data." Line 551

Prior works exist to learn spatio-temporal relations among skeletal data in an unsupervised manner [3,4]. Randomly masking of keypoints to emulate real-world occlusion was explored by Patel et al., 2023 [4].

- "Two of them, GRU and TCN were designed for time-series analysis, but have not been tested specifically on skeleton data." Line 321

Gholami et al. [1] use GRU, and Pavllo et al. [2] use TCN for skeletal data.

We thank the Reviewer for pointing out relevant literature. We added in Methods section (lines 340-341 and 361) references to previous use of GRU, LSTM and TCN for skeleton data analysis. We have also reformulated our statements of novelty (lines 92-95 and 579-582).

Here we want to contrast these works with ours and highlight the novel contributions of our work.

- Gholami et al. [1] and Pavllo et al. [2] are both solving a similar problem: reconstructing 3D poses from tracked 2D keypoints in videos, using GRU and TCN architectures respectively. Both approaches assume near-perfect 2D keypoint tracking, with 3D annotations used only to smooth or correct minor errors rather than as a tool for imputation of missing data. Additionally, the authors test the robustness of their methods with a limited number of camera views, aiming at 3D pose estimation from a single-view. This focus on minimizing camera number seems misaligned with the priorities in lab animal research, where the number of cameras is typically chosen with the goal of achieving precise tracking. Moreover, [1-3] were solely tested on human datasets.
- In [3], the authors explore the use of GAN to forecast 3D human motion from past 3D skeleton poses, including under occlusion. However, their method is not as general as DISK by two aspects: 1/ The authors selected a few strategies when injecting missing data, from randomly choosing the identity of missing joints to masking an entire limb. In all their scenarios, this masking is done independently on each frame and 80% of the sequence is masked. This strategy does not accurately represent real-world imputation tasks where occlusions are often systematic and affect entire body parts over multiple,

consecutive frames. Precisely these types of occlusions are used in DISK training. As the authors of [3] acknowledge, "When the occlusions happen at random, linear interpolation is a good approach, but depending on the nature of the occlusion, we may need a more robust model." For the imputation of occlusion task, they found no improvement of the GAN architecture over the encoder component of their network. 2/ The network training, e.g. specific limb loss or rotation as part of data augmentation, relies on hard-coded limb definition such as head, right arm, right leg, left arm, left leg. **In its current implementation, MotionGAN cannot be applied to species with different anatomy**, such as fish or insects, or even rodents but with different skeleton definition (e.g. tracking of tail or no tracking of the front legs). DISK, in contrast, can be applied to any skeleton type including multi-animal set-ups.

- Optipose [4], like DISK, seeks to develop a generalizable architecture for modeling pose dynamics without requiring biomechanical expertise. By leveraging context models and a self-attention mechanism, the authors frame the task as spatio-temporal refinement of raw 3D keypoints. Their training is bolstered by data augmentation strategies, including rotation, translation, noise addition, and masking.

We added a comparison between DISK, Optipose and Keypoint-MoSeq in Supp. Fig. B3 (reproduced here in Fig. 1). Keypoint-MoSeq ¹ is a recently published method to identify behavior modules from keypoint trajectory data, and has a denoising module to distinguish between noise and real movements. As shown in Fig. 1, Optipose and Keypoint-Moseq capture the overall dynamics of the movements on longer time scales (cf panel a), but fail to impute (cf panels b-f) finer movements. This might be due to the purpose of these two methods being smoothing and denoising rather than precise imputation. We see in the examples displayed in the figure that the denoising process is sometimes erasing real complex movements (cf panel b, cyclic walking pattern on the z axis). Averaged RMSEs on test dataset are 0.60 mm for DISK, 7.24 mm for Keypoint-MoSeq, 7.81 mm for Optipose. **Overall, DISK displays better match with ground truth (cf panels d-f on test dataset) and with trajectories before and after the gap (cf panels a-c).**

C. Strengths of Methodology:

- The authors explored several different neural network architectures: Temporal CNNs, GRU, GCNs, and Transformers.
- They propose a framework that predicts the estimated error in missing skeletal data, which can further be used for thresholding and filtering.
- The authors use a latent space embedding paired with kinematic information to classify behaviors.

Weaknesses in Experimental Design:

1. The paper lacks a quantitative comparison with prior works on benchmark datasets. We recommend that the authors take off-the-shelf pose estimation methods with easily available model checkpoints, such as provided by references [5], [6], and [7], and evaluate whether their method improves the accuracy using the standard pose estimation metrics.

¹Weinreb, C., et al. "Keypoint-MoSeq: parsing behavior by linking point tracking to pose dynamics." Nature Methods 21.7 (2024): 1329-1339.

Figure 1: Comparison of DISK, Optipose and Keypoint-MoSeq (FL2 dataset). a 1-minute-long recording of left knee trajectory. Overall all three methods' outputs match the X and Y values of the original trajectory. On the z axis, movements are within a smaller range as the mouse is exploring a flat surface. In this axis we see discrepancies between original trajectory and output trajectory for Keypoint-MoSeq and especially for OptiPose. **b & c** Two shorter sequences around missing data regions taken from **a**. Inside the gaps, there is no ground truth as these are original gaps in the data. DISK displays better match with dynamics before and after the gaps. **d-f** Three 1-second-long sequences from test data with original ground truth masked as input. **d** right back corresponds to mouse shoulder, **e** left coord to a point in the middle of the mouse back, and **f** to the right knee (see Ignatowska-Jankowska et al. Endocannabinoid system modulation alters fine motor behavior in mice: insights from 3D motion capture. bioRxiv (2023). for details). X, y, and z distances are expressed in mm.

2. The paper does not use standard pose estimation metrics, such as PCK, MPJPE, and P-MPJPE [8,9].

1. The works [5-7] introduce methods for pose estimation from video recordings. In contrast, DISK takes as input 2D or 3D partially tracked trajectories (i.e. with gaps) and imputes missing keypoint data in these trajectories. We therefore cannot provide direct comparison of DISK with the methods [5-7]. Rather than an alternative method to pose estimation methods such as [5-7], DISK is designed to work as a post-processing step of pose estimation. Our implementation available on github allows to use DeepLabCut output files as input into DISK.

2. We thank the reviewer for mentioning the three additional metrics, namely Probability of Correct Keypoint (PCK), mean per-joint position error (MPJPE) and Procrustes aligned mean per joint position error (P-MPJPE). We added PCK and MPJPE in DISK code. We used the same convention for the *reference distance* for PCK as in [4]: "For PCK, we report both PCK@0.05 and PCK@0.1 where *the range is the maximum distance between any pair of ground truth keypoints.*" The P-MPJPE incorporates an additional step to the MPJPE that aligns the predicted and ground-truth positions by applying a non-rigid transformation that minimizes the overall distance between the two sets of points. We designed DISK to generate outputs that can directly be inserted in the original trajectories, without a post-processing alignment step, hence we did not see P-MPJPE being adequate to DISK. We added a paragraph in the Methods section (lines 526-530) that defines the used metrics and provides their formulas. We also plotted the comparison between architectures using MPJPE and PCK@0.01 metric in the Fig. 2 below, and in Supp. Fig. B1. **The new metrics confirm the observations about DISK and other methods we tested in the initial version of the manuscript.**

D. See F about additional experiments and metrics.

E. The use of the metrics mentioned above would better represent the reconstruction quality of the proposed method and show whether its results are competitive with existing techniques and thus provides an improvement that warrants publication in Nature Methods.

We thank the Reviewer for pointing this out. We added PCK and MPJPE as metrics in the DISK code, in the Methods section of the manuscript (lines 526-530), and as the Supp. Fig. B1. These metrics have confirmed and refined the performance assessment of DISK that we presented in the initial version of our manuscript.

F. The quantitative comparison with prior works on benchmark datasets that we recommended above should be relatively easy to do and not require a major revision of the paper. It would result in a few additional paragraphs and new result table.

Of the listed techniques, we added a comparison to OptiPose and to Keypoint-MoSeq as the purpose of these methods match the one of DISK. The other methods listed by the Reviewer, serve a different purpose and have limitations (see point B above) that do not allow for a direct comparison with DISK. The results of the comparison are available in Supp. Fig. B3 (reproduced here in Fig. 1). In summary, DISK outperforms Optipose and Keypoint-MoSeq,

Figure 2: Comparison of different architectures according to the MPJPE (a) and PCK@0.01 (b) metrics. While MPJPE is very close to the originally used RMSE metric, PCK@0.01 displays larger variations between datasets and movement ranges. For example, in DF3D, the fly is maintained at the torso with restrained movements.

especially on finer time scale and finer movements.

The code needs to be fixed so that it can be run as suggested:

- The drive link in the reporting summary does not work.
- The `impute_dataset.py` file for the Rat7M dataset with the pre-trained checkpoint did not work.
- Some crucial keys in "`.hydra/config.yaml`" are missing.

Thank you for your feedback, we fixed the tutorial as available on the drive link https://drive.google.com/file/d/1ycrxt3r9BP6b8N9LYGQ-4y3mdS5ebRGD/view?usp=drive_link. We have corrected the link, `impute_dataset.py` file, and `.yaml` file.

G. References mentioned in this review:

[1] Gholami, Mohsen, et al. "Self-Supervised 3D Human Pose Estimation from Video." *Neurocomputing*, Mar. 2022, <https://doi.org/10.1016/j.neucom.2022.02.076>. Accessed 6

Mar. 2022.

- [2] Pavlo, Dario, et al. “3D Human Pose Estimation in Video with Temporal Convolutions and Semi-Supervised Training.” ArXiv.org, 29 Mar. 2019, arxiv.org/abs/1811.11742.
- [3] Ruiz, Alejandro Hernandez, et al. “Human Motion Prediction via Spatio-Temporal Inpainting.” ArXiv.org, 28 Oct. 2019, arxiv.org/abs/1812.05478. Accessed 30 May 2024.
- [4] Patel, M., Gu, Y., Carstensen, L.C. et al. Animal Pose Tracking: 3D Multimodal Dataset and Token-based Pose Optimization. Int J Comput Vis 131, 514–530 (2023). <https://doi.org/10.1007/s11263-022-01714-5>
- [5] Iskakov, K., Burkov, E., Lempitsky, V., & Malkov, Y. (2019). Learnable Triangulation of Human Pose. https://openaccess.thecvf.com/content_ICCV_2019/html/Iskakov_Learnable_Triangulation_of_Human_Pose_ICCV_2019_paper.html
- [6] He, Y., Yan, R., Fragkiadaki, K., & Yu, S. I. (2020). Epipolar Transformers. https://openaccess.thecvf.com/content_CVPR_2020/html/He_Epipolar_Transformers_CVPR_2020_paper.html
- [7] Mathis, A., Mamidanna, P., Cury, K.M. et al. DeepLabCut: markerless pose estimation of user-defined body parts with deep learning. Nat Neurosci 21, 1281–1289 (2018). <https://doi.org/10.1038/s41593-018-0209-y>
- [8] Mehta, D., Rhodin, H., Casas, D., Fua, P., Sotnychenko, O., Xu, W., & Theobalt, C. (2017). Monocular 3D Human Pose Estimation In The Wild Using Improved CNN Supervision. <https://vcai.mpi-inf.mpg.de/3dhp-dataset/>
- [9] MODEC: Multimodal Decomposable Models for Human Pose Estimation. (2013, June 1). IEEE Conference Publication — IEEE Xplore. <https://ieeexplore.ieee.org/document/6619315>

H. Clarity and context are good.

Reviewer #1 (Remarks on code availability):

The drive link in the reporting summary does not work. The `impute_dataset.py` file for the Rat7M dataset with the pre-trained checkpoint did not work. Some crucial keys in `hydra/config.yaml` are missing.

Thank you for your feedback, we fixed the tutorial as available on the drive link https://drive.google.com/file/d/1ycrxt3r9BP6b8N9LYGQ-4y3mdS5ebRGD/view?usp=drive_link. We have corrected the link, `impute_dataset.py` file, and `.yaml` file.

Reviewer #2 (Remarks to the Author):

This study by Rose et al aims to address missing data in kinematic tracking and pose estimation methods and offers a method for imputing 2D and 3D tracking data as well as assessing the accuracy of imputation in order to prepare data for downstream analyses. There is a growing need for methods to accurately impute missing and spurious skeleton tracking data due to the growing use of markerless tracking methods in animal behavior studies, and a simple tool that could be run on any kinematic data to clean up missing keypoints would be of great use. However, I have several reservations about the method and claims presented here.

Major Issues:

1. While the claim here is that the keypoint dependencies and dynamics are used to impute missing tracking data, why are the snippets for training limited to ~ 60 frames?

Indeed in the main text, we only show results on samples of 60 frames. However in the supplementary figure A12 (also shown in this response in Fig. 3 page 9), we test the influence of several data-related parameters on DISK performance: the training dataset size (panels a and b) and the length of input sequence from 60 to 480 (panels c-f).

- Panel c shows that considering longer sequences improves the imputation for the same gap probability distribution, while panel d shows the increase of computational time as the sequence size increases. We can see that the training time of transformer models increases with the input size, while the training time of GRU models decreases. Indeed, the transformer model size scales with input size, as we use one token per keypoint per timeframe.
- Panels e and f refine this by showing the RMSE as a function of the length of the gaps. Panel f is a zoom of panel e without linear interpolation. We can see that the improvement spans across different gap lengths and is not limited to small or long gaps.

These tests were performed on the 2-Fish dataset, our biggest dataset (cf Table 1 in main text). Indeed, the size of training set in terms of number of sequences is an important parameter for DISK performance. As a rule of thumb for the user, we recommend to start with a sample length of 30 to 60 and only increase the sample length if it does not deplete the training set excessively. As the training set is built from intact portions of the original data, increasing the sequence length will decrease the number of sequences that can be used in training. From our experiments, we found that around 2,000 training sequences is sufficient for obtaining good accuracy with DISK. These observations are described in lines 612-616 in the manuscript.

2. Does this differ if imaging at a higher frame rate?

The datasets that we tested in the manuscript have different frame rates (cf Table 1 in the manuscript). **Our results show that the frame rate does not correlate with imputation accuracy.** For example, the datasets FL2 and Human have a frame rate different by a factor of 3, however show a comparable imputation accuracy. DISK code has an option to downsample the data when creating the DISK dataset formats.

Figure 3: DISK performance dependence on dataset size and input length (2-Fish dataset) **a** Performance when decreasing the size of the dataset by subsampling. The size of the dataset influences the performance: the more input sequences the better. However a division factor of 10 of the original dataset gives still satisfactory test RMSE under 0.04. **b** Training time for different size of subsampled datasets (batch size = 32) for GRU and transformer. **c** Test RMSE for GRU, transformer, and linear interpolation. An increased input sequence length while keeping the stride constant (same dataset size) show increased performance. **d** Training size for different input length (dataset size kept constant). Transformer has a big memory footprint and the model size is exceeding the 32GB of memory of a V100 GPU at length 480. On the other side, GRU training time is kept constant for all tested input length. **e - f** Test RMSE with respect to gap length. Increased input length shows lower error, even on shorter gaps. Increased input size seems to provide a more complete picture of the dynamics and help the imputation. **f** is a zoom of **e** without the linear interpolation line.

Figure 4: Mean RMSE for the same trained DISK model and different values of parameters `pad_before` and `pad_after` set at test time. Linear interpolation (left matrix) can only be computed when at least one data point at the beginning and one data point at the end of the sequence are available (corresponding to `pad_before` > 0 and `pad_after` > 0). For values where linear interpolation can be computed, there is little influence of `pad_before` and `pad_after`, and variations can be considered as stochastic. For DISK (right matrix), we see that `pad_before` > 0 yields lower errors. The dynamics of a sample is usually well imputed, but the exact starting coordinates help reducing the error.

3. What happens when dropouts are longer than this but could easily be restored with other methods (even simple interpolation), for example when an animal is still but one marker or keypoint is not visible.

Gaps longer than the DISK input sequence will a priori not be imputed. Hence some gaps will remain after imputation (see for example Figure 7c in the manuscript). **The user can overwrite this default behavior** by changing the `pad_before` and `pad_after` parameters and set them to 0. These parameters control the minimal number of non-missing datapoints before and after a gap for it to be imputed by DISK. When they are both set to 0, sequences with keypoints with all values missing will also be imputed by DISK. Based on our experience, setting `pad_after` to 0 has a minor effect in the performance (but is required to be > 1 if we want to compare with linear interpolation for a given sample). In contrast, `pad_before` helps the model performance by providing the starting value of the sequence (cf Fig. 4).

4. One major issue with tracking data (especially in marker-based tracking) is the tendency to switch keypoints (even more so in social tracking), have the authors done anything to address how to identify and correct these flips?

When two markers come close to each other they might be switched for a few frames, however it is very challenging to automatically detect such an event in trajectory data. Indeed because of their small distance the "jump" from one to the other might appear as a plausible fast movement. In markerless data, one could go back to the video frames when they are available to verify if a switch happened. In marker-based tracking, no video is generated and only the keypoint trajectories are available. Acknowledging this difficulty, we added as data augmentation in the training a *switch* data augmentation that switches part of the sequence between two randomly chosen keypoints. This additional data augmentation aims at making

DISK more robust to potential keypoint switches in the real data. We tested it by comparing two DISK proba models: one trained with the *switch* data augmentation and one without. We compared the imputation error made by both models on a held-out test dataset where we artificially switched two keypoints in each sample. We can see in Fig. 5 (added as Supp. Fig. B4 in the manuscript) that DISK proba model trained with our *switch* data augmentation is better at correcting switches at test time.

5. The strongly worded claim that ‘no general missing data imputation method has been developed for skeleton data’ sends one looking for counterexamples - would something like the method developed here for one of the manuscript’s use cases (**MarkerBasedImputation**) not qualify as it is not general enough? In that case could a direct comparison be worthwhile to show how this method performs against a method designed with a particular dataset in mind?

We thank the Reviewer to bring the github code base github.com/diegoaldarondo/MarkerBasedImputation to our attention. This work is indeed very close to our approach, though was not, to our knowledge, published or used in published results. In this code base, LSTM and WaveNet architectures are available. LSTM is close to our tested GRU, and WaveNet is a type of TCN with dilated and causal convolutions. Our TCN has dilated but non causal convolutions as we wanted to use the information before and after the gap. The code base is at a developer stage, with the last contribution dated from five years ago, and some parameters are hard-coded such as the number of frames necessary (9 frames) before a gap. As such this method is not usable in its current stage of development and has some important limitations to apply it to various skeleton datasets that we used in our manuscript.

Additionally we rephrased the claim about our novelty in lines 92-95 and 579-582..

6. Accuracy metrics for interpolation will always be biased to common postures that might be more easily tracked - I would like to see how the algorithm performs on difficult/ less common postures that might actually limit downstream analyses. Given that some of these datasets are annotated for behavior - has there been any attempt to test this method on a subset of behaviors during which tracking fails more commonly? Introducing missing data at random might bias the results to appearing more accurate than they really are for given tricky subsets of behavior.

During unconstrained behavior, it is likely to encounter pose sequences repeated at varying frequencies. We have access to action labels in two of the collected datasets: the Human MoCap dataset and the 2-Mice-2D dataset. In the human datasets, 8 action classes are available in different proportions : walk (19%), wash (5%), run (4%), jump (4%), ”animal behavior” (4%), dance (3%), step (3%), climb (1%) and unknown (57%). In the 2-Mice-2D dataset 4 classes are available: attack (3%), mount (5%), investigation (29%) and other (63%). In the human dataset, the least represented class, climb, is not associated with the highest error, and the most common class, walk, with the lowest error (cf Fig. 6 below). In the 2-Mice-2D dataset, attack and mount classes are present in comparable proportions but attack is associated with the highest error. Overall, the confidence interval is quite large and the accuracy overlaps between classes.

In addition to differences between actions, we could hypothesize that in multi-animal set-ups, when the two animals are close to each other, tracking would be more difficult. Yet on

Figure 5: Imputation of switched keypoints. (2-Mice-2D dataset) **a, b, c** Three examples with switched keypoints at test time. The blue trajectory represents the ground truth; the dashed grey line is the sample with the switched keypoints given as input to the DISK proba models; the straight grey line is the linear interpolation of the gap. In yellow are shown the trajectory and uncertainty for the DISK proba model trained with the switch data augmentation. In red the DISK model trained with the switch data augmentation. **d** Averaged RMSE, PCK@0.01 and MPJPE for linear interpolation and the two DISK proba models. Using switching as data augmentation improves the model's accuracy in case of switched keypoints. All results here are expressed in normalized units (see Method paragraph 3.4 in the manuscript).

Figure 6: Average RMSE by action category on **a** 2-Mice-2D and **b** Human datasets.

the fish dataset (Figure 4c in the manuscript), we show that the imputation error increases with the distance between fish, which is counter intuitive. Additionally, we see that, in the case when all the keypoints of at least one fish are missing (pink, purple and gray lines in Figure 4c in the manuscript) the imputation error also is the smallest when the fish are close to each other. Typically video-based tracking algorithms fail in such situations of close proximity, while DISK takes advantage of the correlation between interacting animals and remains robust. **We therefore do not find specific action classes which are more difficult to impute based on their appearance frequency, however observe that in 2-animal recordings, poses of animals in close proximity to one another are easier to impute.** This effect is described in the manuscript on pages 11-12.

7. There is a trade-off between recovering missing data via interpolation and training better models that incorporate skeletal information (especially for 3D data with good calibration) - a discussion on this, limitations of this method, and future directions that could incorporate this type interpolation with tracking seems warranted.

We added following remarks to the manuscript:

- In the Introduction (lines 117-124), "While in markerless tracking, extensive manual annotations can allow to obtain high coverage and tracking accuracy, yet, the lighting conditions or angle of view can make some frames challenging or even impossible to annotate with high confidence. DISK's imputation strategy is based on learning of the motion dynamics and dependencies between keypoints, which can allow to overcome the limitations of keypoint visibility. Additionally, manual annotations is a time-consuming task, whereas DISK handles keypoint trajectory data in an automated and annotation-free manner.
- In the Discussion (lines 624-630), "Pose estimation algorithms like DeepLabCut and SLEAP can be trained to predict the complete set of an animal's coordinates under good lighting and viewpoint conditions, and incorporating additional annotations can enhance the precision of these models. However, DISK can alleviate the annotation effort required to achieve comparable coverage and tracking accuracy. As more behavioral datasets are produced and analyzed, the optimal approach to integrating these

methods for accurate and efficient results will become clearer.”

Minor Issues:

- 115 - A keypoint tracking network can be forced to have no missing values - and can always be retrained to be more accurate - so the fear of losing data and needing to take new recordings when the videos exist seems unfounded.

While additional manual annotations are a priori feasible, in certain situations, the lighting conditions or angle of view can make it challenging to annotate with high confidence all keypoints. Similar reasons can result in automatic tracking associated with poor confidence. While keypoint tracking tracks each keypoint separately, DISK employs a different strategy and is based on learning the tracking dynamics and correlation between keypoints. Additionally, manual annotation is a laborious, time-consuming task, while using DISK could save researchers’ time by its automated and annotation-free design.

We precised this in the manuscript in the Introduction (lines 117-124) and in the Discussion (lines 624-630).

- Figure 1 - Not sure why both panels a and c exist here, also for the Rat7M dataset - does this refer to the total fraction of frames with missing keypoints or the total fraction of missing keypoints? Since the authors performed their own interpolation, how does the method here compare?

In Figure 1 of the manuscript, panel a refers to the application case of DISK, while panel c displays the training strategy. For panels d-i, the percent originally missing refers to the percentage of timepoints with at least one missing keypoint in the original files and original sampling frequency. We have now reformulated the caption of Figure 1 to include this information. In the Rat7M dataset from Dunn et al., we use directly the motion capture data, which is not further interpolated, but in their work as ground truth to build the DANNCE network. The DANNCE approach estimates 3D pose directly from the 2D video frames. As this is a different task than imputation, we cannot compare here to the DANNCE performance.

- Figure 2 - How is RMSE calculated for 2D data? The individual panels here seem unnecessary - they could be combined into a stacked plot and include a much longer time window to give a sense of how the interpolation actually looks over time. In b-i, what is the ground truth for any of the tracks (and does b really need 7 examples of tracks with no missing data and no interpolation)?

RMSE calculated on 2D data, the same as 3D, average over the spatial dimensions (x, y) or (x, y, z), respectively:

$$RMSE_{2D} = \sqrt{\sum_{t=1}^T (\hat{x}_t - x_t^{GT})^2 + (\hat{y}_t - y_t^{GT})^2}$$

For more clarity, we added the formulas of the metrics in the Methods section (lines 526-530)

Panel b in the Figure 2 of the manuscript shows the missing keypoint as well as other non missing keypoints. Non-missing keypoints' trajectories are displayed to illustrate the information available to DISK during imputation. Similarly, panel c only displays the keypoint with missing data and two other keypoints, one on the same arm as the missing one and one on the opposite arm. All examples shown in Figure 2 contain one missing keypoint each. We have added more explanations to the figure caption and would like to keep those panels for the reasons listed above.

To give a sense of the results of imputation on a longer time window, the reader can refer to Figure 7c and Supp. Fig. B3 of the manuscript (reproduced here in Fig. 1).

- Figures 5 and 6 seem like an incomplete departure to a different story altogether - it seems obvious that latent spaces encoding posture and dynamics would segregate based on type of behavior being performed, is the point here that the latents encode something about posture and dynamics? What's going on with the 'walk' skeleton in Figure 5a? Either way, is seems like a distraction from the main point.

Figures 5 and 6 are indeed an outlook into a research direction that should be explored in more detail in the future. The reason for including these plots in the manuscript is to hint in that direction and show the potential of self-supervised learning of representations of behavior. Such learning, based on masking as well as contrastive examples is broadly used in computer vision and language processing, however has not been yet fully explored in behavior analysis. We additionally consider that these visualizations help to add interpretability to our deep learning model. We discuss these points in the manuscript lines 591-600.

- 293 - the language here suggests that a larger number of steps detected is good - but why not use an artificial dataset that has a true 'correct' number of steps as well as dynamics and then perform the test on that? If this was done in 3D then a true distance to the target could be calculated at every point in time.

We feel that validation on real data is more relevant compared to an artificially generated step trajectories. Detecting more steps is not necessarily good, we could imagine fake steps to be detected. That is why we validated our detection against manual counting of steps per recordings (cf Figure 7d in the manuscript) and demonstrate that the imputed results are close to this manually generated ground truth (cf Figure 7b in the manuscript).

Answers to Reviewers' comments

Reviewer #1

Remarks to the Author:

The paper discusses a method for learning spatio-temporal relations among the keypoints of the skeleton by augmenting the training dataset by randomly masking a subset of keypoints and thus emulating occlusion in real-world data. The authors conducted experiments on diverse animal subjects (humans, mice, rats, flies, and fish). Furthermore, they analyze the latent embedding of their model to classify behaviors.

- The authors explored several different neural network architectures: Temporal CNNs, GRU, GCNs, and Transformers.
- They propose a framework that predicts the estimated error in missing skeletal data, which can further be used for thresholding and filtering.
- The authors use a latent space embedding paired with kinematic information to classify behaviors.
- The paper uses well-defined standard pose estimation metrics.
- The authors demonstrate their method on Mouse FL2 dataset by predicting missing data; which made their results closer to manual counts performed by experts.

The code issue has been fixed. The provided notebook is easy to follow and covers all relevant use cases. Training a model and running it in inference mode seem straightforward. The code is reasonably well-documented and easy to follow. The authors have also provided an FAQ that covers what the hyperparameters do and how to change them.

We heartly thank the reviewer for their positive feedback.

Reviewer #2

The edited manuscript by Rose et al addressed some of my comments and overall presents a method that may be useful in imputing tracking data from behavioral studies. The main issues previously I raised about the proposed method were longer dropouts and missing data, keypoint switches, and a lack of acknowledgment for previously published methods. DISK is suitable for many types of skeletons, including multi-animal set-ups, and the authors have demonstrated improvement in tracking using DISK in several use cases. For this revision, the authors have added comparison to other methods, more accuracy metrics, and some additional experiments testing keypoint switching. The comparison to Kp-Moseq and Optipose, which I had not originally considered, raised more questions about why the others methods have such poor outputs outside of interpolation gaps.

We thank the Reviewer for thoroughly reading our manuscript and for their follow-up comments.

Optipose and Keypoint-MoSeq are designed for a different purpose than DISK. While the goal of Keypoint-MoSeq and Optipose is to denoise input data, smoothen it, remove the outliers, etc., DISK is not designed to modify existing data but to impute the data that is missing. Keypoint-MoSeq and Optipose outputs are a modified version of the input data, which is perceived as an error and referred by the Reviewer as "poor outputs outside of interpolation gaps". In the example shown (see panels h - i of the updated Fig. 2 in the manuscript, reproduced here as Fig. 1), OptiPose and Keypoint-MoSeq follow the global trajectories, and modify jittery and fast movements, as seen when zooming in the temporal and spatial dimensions (see in particular z axis expressed in mm). This example illustrates the difference in the methods' objectives: Keypoint-MoSeq and Optipose were designed to perform precisely this smoothing and denoising task, while DISK was designed to fill in the missing data.

For this reason, it is challenging to compare original results reported in the Optipose or Keypoint-MoSeq studies, even on the same datasets, e.g. Rat7M. In the Optipose study, the authors test the correcting power of their method by corrupting motion capture data from the Rat7M considered as ground truth by adding noise and missing data to it. The comparison with DISK is difficult, as DISK does not modify the input where there is no gap. The results reported in the paper relate to the entire trace of keypoints while DISK will be only evaluated in created gaps. For a fair comparison, we therefore reported the metrics – root mean squared error (RMSE), mean per joint position error (MPJPE), and percent of correct keypoints (PCK@0.01) – only *on the missing data* and did not include metrics outside of the gaps. We completed the previous results by reporting summary metrics on all seven datasets (see added panels Fig. 2 b / Fig. 1b and Supp. Fig. A1 / Fig. 2 c-d in the manuscript / in this rebuttal resp.), and metrics distributions with respect to gap length and number of missing keypoints (cf added panels Fig. 2 g / Fig. 1 g and updated Supp. Fig. A3 / Fig. 3 in the manuscript / in this rebuttal resp.). We discuss these results in the manuscript lines 169-176 and 186-195, and later in this rebuttal (page 5).

Additionally, we share our code for running the comparison to enforce the reproducibility of this analysis in a separate github repository DISK_paper_code.

[1] Patel, M., Gu, Y., Carstensen, L.C. et al. Animal Pose Tracking: 3D Multimodal Dataset and Token-based Pose Optimization. Int J Comput Vis 131, 514–530 (2023). 10.1007/s11263-022-01714-5

The test of switch keypoints shows that accuracy is improved and that the uncertainty of the model might be useful in a more comprehensive switching detector and interpolator. However, as with most of the quantification in the original manuscript and the rebuttal, I would like to see summary statistics of switch types (within single animal, mutli-animal, across species) across many lengths and types of switches instead of just a few traces.

Indeed, our additional experiment in the first revision showed that DISK does correct switched keypoints when trained with an additional data augmentation module. While we demonstrated this in the scenario of 2 randomly chosen switched keypoints, we would like to underline that there are many possible keypoint switch scenarios in interplay with the missing data, in particular across all animal skeletons used in our manuscript. Which keypoints, how many of them, and when should these keypoints be switched relative to the missing data is an open question, leads to many possible training scenarios and makes the interpretation of results difficult. In contrast to missing data, inferring from the existing data which keypoints got switched during animal skeleton tracking, with what frequency and for how long is not trivial. This hinders simulating such realistic switches in our training data.

Having that in mind, we extended the switched keypoint analysis to the remaining datasets (including different skeletons), i.e. trained DISK models with this additional data augmentation module. We reported summary statistics (PCK, RMSE, MPJPE; see new Supp. Fig. A6 c, reproduced here as Fig. 5, and new Supp. Fig. A7, reproduced here as Fig. 6). The performance of both DISK models - trained with and without the switch module - on samples without switched keypoints is similar (cf Fig. 5c with ‘swap=False’, MPJPE of 6.28 mm for DISK with switch module vs 6.08 mm for DISK without switch module, 2-Mice-2D dataset). However, the additional switch module improves the imputation of samples containing switched keypoints (cf Fig. 5c barplots with ‘swap=True’, MPJPE of 9.47 for DISK with switch module vs 19.86 mm for DISK without switch module, 2-Mice-2D dataset). Across datasets, we observe the same pattern with the DISK model with switch module having an on-par performance on the non-switched samples, while having improved performance on the switched samples. The least improvement is observed on the DF3D dataset, with high performance of both models (PCK@0.01 of almost 1), while the most improvement is seen in Fish (MPJPE of 0.079 mm for the non-switched samples, and 0.13 vs 1.14 mm for

the DISK with and DISK without switch module respectively).

Furthermore, we reported PCK@0.01, as a measure of the percentage of corrected traverses, with respect to the average distance between the two switched keypoints during the switch (cf panels f & i of Supp. Fig. A6 / Fig. 5, as well as new Supp. Fig. A8 / Fig. 7 (right column) in the manuscript / in this rebuttal resp.). Globally across datasets, DISK does not show a tendency for a higher or lower error when the two points are separated by a short distance rather than a large distance. Yet, switches due to tracking errors occur more frequently when the distance between the keypoints is small as their tracking becomes more challenging. DISK therefore is able to correct these close-by keypoint switches in real conditions, a very useful feature given that this type of switches occur more frequently as the result of tracking errors.

We reported separately the metrics for each pair of keypoint identities being switched (see Supp. Fig. A6 / Fig. 5 d & g and first columns of Supp. Fig. A8 / Fig. 7 in the manuscript / the rebuttal resp.). In the 2-Fish dataset, we see that the imputation error is larger (lower PCK) when the two switched keypoints belong to different animals, independent of their precise identity (e.g., tail, head or pec fin in the case of fish), potentially due to a lower correlation between the dynamics of the two switched keypoints. However this effect is not seen in the 2-Mice-2D dataset. For the single animal datasets, no overall trend was observed: DISK seems agnostic to the identity of switched keypoints.

When looking at the percentage of corrected traces with respect with the gap length and the swap length (see Supp. Fig. A6 / Fig. 5 e & h and middle column of Supp. Fig. A8 / Fig. 7 in the manuscript / in this rebuttal resp.), we see a decrease of DISK performance when both the gap length and the swap length increase.

As a conclusion, the switch module improves DISK’s robustness to switched keypoints, independent of the identity of keypoints, but with a lower corrective power as the length of the gap and/or the length of the switching increases. We discuss these results lines 214-222 and in the discussion lines 666-668 in the revised manuscript.

I appreciate the authors have produced a much more generalizable method, but the code base github.com/diegolaldarondo/MarkerBasedImputation was published as part of the CAPTURE paper, from which this manuscript uses data (Rat7M)! Arguing that it was never published or used in published results is misguided, unless there is something I am missing here. The authors also argue that the Rat7M data was not further interpolated in the rebuttal - I think this is directly addressed in the CAPTURE paper and supplement.

We apologize for what we believe is a genuine misunderstanding. To the best of our knowledge, the Rat7M dataset was published as part of the "DANNCE" paper [2] and we referenced it as such in the manuscript. In the supplementary material of the "DANNCE" paper, the dataset is described as follows: "We collected 1,164,343 timepoints of $1,320 \times 1,048$ pixels color video data at 30 Hz from 6 synchronized cameras (Flea3 FL3-U3-13S2C, Point Grey) at 30 viewpoints overall. This yielded a total of 6,986,058 images, together with motion capture data (12 cameras) for 20 landmarks, from 6 different rats with affixed retroreflective markers. Animals were lightly shaved and equipped with a headcap to accommodate neural recordings for separate work. *Due to occlusion, sometimes the full set of markers was not tracked in a frame; in these cases, we left untracked markers as missing data points and did not impute their positions.*" This was the reason we assumed the keypoints in this dataset have not been interpolated. The "CAPTURE" paper [3] *does* mention data imputation and the "MarkerBasedImputation" github repository, but does not refer to the Rat7M dataset. Concurrently, the "MarkerBasedImputation" github repository does not link to neither of the papers, which is why we were not aware that it has been published. Based on these publications, we concluded that the online Rat7M dataset was not imputed prior to its upload.

Previously, we ran into difficulties to reproduce the imputation directly from the github repository

”MarkerBasedImputation”, partly because it is lacking compatibility with the recent versions of packages such as tensorflow and the GPU-backend it requires, partly because it does not detail how the input files should be formatted. With the additional information from the paper method section [3], we have now reproduced the data processing and imputation pipeline (additional github repository DISK_paper_code) and added it to the comparison with DISK (cf updated Fig. 2 / Fig. 1 b-e, g and updated Supp. Fig. A1 and A3 / Fig. 2 and Fig. 3 in the manuscript / in this rebuttal resp.). DISK displays the lowest error across methods and datasets, by a factor of 2 or more. While the ranking of other methods vary from dataset to dataset, DISK is consistently the best method for gap imputation.

[2] ”DANNCE” paper: Dunn, T., Marshall, J., Severson, K., Aldarondo, D., Hildebrand, D., Chetih, S., Wang, W., Gellis, A., Carlson, D., Aronov, D., Freiwald, W., Wang, F., Ölveczky, B. (2021). Geometric deep learning enables 3D kinematic profiling across species and environments. Nature Methods. 18. 1-10. 10.1038/s41592-021-01106-6.

[3] ”CAPTURE” paper: Marshall, J., Aldarondo, D., Dunn, T., Wang, W., Berman, G., Ölveczky, B. (2021) Continuous Whole-Body 3D Kinematic Recordings across the Rodent Behavioral Repertoire, Neuron. 109.3. 420-437.e8, ISSN 0896-6273, 10.1016/j.neuron.2020.11.016.

I am also still very confused by the argument that ‘validation on real data is more relevant compared to artificially generated step trajectories’ as an argument against generating artificial, corrupting it, and then testing DISK on different types of dynamics and fraction of missing data. This would allow for a broader understanding of the limits of the types of movements DISK is good at capturing and interpolating, instead of limiting imputation results to a single type of behavioral difference in a very particular test case.

We understand that the Reviewer is pointing here to the argument in the previous revision about the step analysis presented in Figure 7 in the manuscript. This analysis involves the application of DISK to step detection in freely moving mice, and the comparison of the step dynamics between two groups of mice. Importantly, the work presented in Figures 2-6 was performed to help the understanding of the functioning and limits of DISK, while the purpose of the analysis in Figure 7 was to demonstrate the practical use of DISK in a behavior analysis question that a potential user of DISK might encounter. We would also like to underline that we have chosen step detection for this analysis as a very challenging behavior to detect. In the dataset shown in Figure 7, the mouse is freely moving on a 2D surface, *without being trained to continuously perform forward locomotion*. Hence the steps can be irregular and point to any direction, and leg movements can easily be mis-detected as a step. The imputation and the detection methods need to be sensitive enough to detect all steps and robust to false positives, such as limb movements outside of a full step cycle and forward locomotion.

We would like to address the Reviewer’s comment about ”limiting imputation results to a single type of behavioral difference in a very particular test case”, by pointing the Reviewer to the analyses we presented in Figures 2-6. In these analyses: we tested DISK on 7 datasets, including 2D and 3D, 1 and 2 animals (see Fig. 2 and Supp. Fig. A1); we compared different number of simultaneous missing keypoints (cf Fig. 4) and their switches; we reported the performance of DISK with respect to the length of the gaps (see Fig. 1j and Supp. Fig. A3); we also analyzed DISK performance in the 2-Fish dataset depending on the distance between fish and the number of missing keypoints.

We have chosen real data with actually missing keypoints for the method training as we considered such data as the most relevant for a data-based machine learning method. To artificially generate the full breadth and complexity of natural behavior, without bias, across the range of species that we consider in our manuscript is a remarkably challenging task. To answer the Reviewer’s comment we have searched for artificial behavior datasets and methods for their generation. Methods [4 -

[6] allow for generating artificial multi-keypoint poses. However these approaches allow to generate single images or scenes, as part of pose detection pipelines but not pose sequences. VAME (deep variational embeddings of animal motion) [7] is based on a variational auto-encoder and is by design a generative method. Though it would be potentially feasible to generate short sequences (~ 30 frames-long) with VAME, it has never been used for this purpose. To parametrize VAME in order to generate diverse and realistic sequences requires therefore extensive development and testing which we feel is out of scope of our current work. Last year, a *virtual rodent* was described [8], for which an artificial neural network was trained to generate realistic rat movements inside the MuJoCo simulation environment. To the best of our knowledge, the trained model was however not published, and the code repository includes two folders with not self-explanatory names, lacking documentation or information on how to reproduce the training (see code repository). While the artificial dataset is available upon request, as stated in the paper, without access to the generative algorithm, it would be impossible to control the generated behaviors and corrupt it, as suggested by the Reviewer. We additionally found several biomechanical models [9 - 12] that allow to generate detailed and realistic movements, however such approaches are limited to generation of very specific actions, such as forward gait.

In this revision we analyzed in more detail DISK’s performance per keypoint in all datasets (see Supp. Fig. A2 / Fig. 4 in the manuscript / in this rebuttal resp.), and per action categories in the two datasets with action labels – 2-Mice-2D and Human Mocap – and according to the periodicity and the velocity of the movement (see Supp. Fig. A4 / Fig. 8 in the manuscript / in this rebuttal resp.).

The per keypoint analysis suggests that across datasets the extreme skeleton points (limb ends, head, tail) are less accurately imputed compared to core skeleton points (e.g. spine, hips, or pectoral fin). This result follows the intuition that keypoints with highly correlated neighbors are easier to impute than the end nodes of the skeleton graph. On the 2-Fish dataset, the same performance is associated with each keypoint on both animals, contrary to the 2-Mice-2D dataset where this symmetry is not observed for keypoint 0, 3, and 6. DISK, as a data-driven method, performs slightly differently on each dataset, depending on the range of dynamics represented in the data, the potential difference between left and right side or one and the other animal. We added this to the revised manuscript in lines 196-204.

Then, we inspected the accuracy depending on the action class and overall movement periodicity in the 2-Mice-2D and Human datasets. The results suggest that the imputation error tends to increase with average movement speed and gap length, independently of the action class (cf panels c, e, h, j of Supp. Fig. A4 / Fig. 8 in the manuscript / in this rebuttal resp.). For the periodicity, we also see a slight error increase independent of the action class (cf panels d & i in Fig. 8). There seem to be differences in performance across action classes, with a higher error in the "attack" class in the 2-Mice-2D dataset which is also the least represented (2.8% of the overall data), and in the "run" and "dance" class, representing resp. 4.3% and 3.1% of the overall data. Notably, the least represented classes of the Human dataset is "Climb" with 1.3% followed by "Step" with 2.7%, which do not display the highest errors. This suggests that, although the abundance and diversity of training examples of each *action type* is important for DISK’s performance, action categories might not be the best indicator of DISK’s performance on each inferred sample. We discuss this in the revised manuscript in lines 205-207.

[4] Han, Y., Chen, K., Wang, Y., Liu, W., Wang, Z., Wang, X., Han, C., Liao, J., Huang, K., Cai, S., Huang, Y., Wang, N., Li, J., Song, Y., Li, J., Wang, G., Wang, L., Zhang, Y., Wei, P. (2024) Multi-animal 3D social pose estimation, identification and behaviour embedding with a few-shot learning framework. Nat Mach Intell 6, 48–61. 10.1038/s42256-023-00776-5

[5] Plum, F., Bulla, R., Beck, H.K., Imirzian, N., Labonte, D. (2023) replicAnt: a pipeline for generating annotated images of animals in complex environments using Unreal Engine. Nat Commun 14, 7195 (2023). 10.1038/s41467-023-42898-9

[6] Lyu, J., Zhu, T., Gu, Y., Lin, L., Cheng, P., Liu, Y., Tang, X. and An, L. (2024) AniMer: Animal

- Pose and Shape Estimation Using Family Aware Transformer. arXiv preprint arXiv:2412.00837
- [7] Luxem, K., Mocellin, P., Fuhrmann, F., Kürsch, J., Miller, S.R., Palop, J.J., Remy, S. and Bauer, P. (2022) Identifying behavioral structure from deep variational embeddings of animal motion. *Communications Biology*, 5(1), p.1267. 10.1038/s42003-022-04080-7
- [8] Aldarondo, D., Merel, J., Marshall, J.D. et al. (2024) A virtual rodent predicts the structure of neural activity across behaviours. *Nature* 632, 594–602. 10.1038/s41586-024-07633-4
- [9] Falisse, A., Serrancolí, G., Dembia, C. L., Gillis, J., Jonkers, I., De Groote F. (2019) Rapid predictive simulations with complex musculoskeletal models suggest that diverse healthy and pathological human gaits can emerge from similar control strategies *J. R. Soc. Interface*.1620190402 10.1098/rsif.2019.0402
- [10] Ezati, M., Ghannadi, B., McPhee, J. (2019) A review of simulation methods for human movement dynamics with emphasis on gait. *Multibody Syst Dyn* 47, 265–292. 10.1007/s11044-019-09685-1
- [11] Gilmer, J.I., Coltman, S.K., Cuenu, G., Hutchinson, J.R., Huber, D., Person, A.L., Al Borno, M. (2024) A novel biomechanical model of the mouse forelimb predicts muscle activity in optimal control simulations of reaching movements. *bioRxiv* 2024.09.05.611289; 10.1101/2024.09.05.611289
- [12] Charles, J.P., Cappellari, O., Hutchinson J.R. (2018) A Dynamic Simulation of Musculoskeletal Function in the Mouse Hindlimb During Trotting Locomotion. *Front. Bioeng. Biotechnol.* 6:61. 10.3389/fbioe.2018.00061

In the same vein, while the result that the multi-animal interpolation in the fish data is better when they are close - this is a result that contradicts most multi-animal tracking wisdom and could also be properly explored by generating fake multi-animal trajectories and testing the method on balanced sets of deleted data.

DISK is indeed not a tracking method. Our method uses as input data the result of tracking and pose detection methods and imputes the parts of the data that are missing. In the case of 2 fish performing a fight, poses of the two animals are closely dependent from one another which is the reason that DISK’s performance improves in this scenario. While animals in close proximity are a challenge for tracking and detection methods, DISK takes advantage of the interdependence of the two animal poses to improve its imputation. Previous work [13] has shown that there is high correlation (and even synchronization) across individuals during free social behavior. Such correlation facilitates the imputation task of DISK. We explained this thoroughly in the revised manuscript (lines 286-290).

[13] Klibaite U, Shaevitz JW (2020) Paired fruit flies synchronize behavior: Uncovering social interactions in *Drosophila melanogaster*. *PLOS Computational Biology* 16(10): e1008230. 10.1371/journal.pcbi.1008230

Overall, the original manuscript and rebuttal have convinced me that DISK could be an incremental advance in imputation, but that the comparison to and scope within the broader context of marker-based and marker-less tracking is limited without additional demonstration of ease of use and accuracy.

We hope to have convinced the Reviewer with the additional analyses and responses presented above.

Figure 1: DISK and other imputation methods' performance across datasets.

Figure 1: **a** RMSE on all datasets for all tested architectures. "proba" refers to a modification of the transformer and GRU architecture to additionally output a confidence interval alongside to the imputed values (see manuscript). Corresponding barplots for mean per-joint position error (MPJPE) and Probability of Correct Keypoint (PCK) metrics are available in Fig. 2. **b** RMSE on all datasets for DISK and other tested methods, MarkerBasedImputation (MBI), Keypoint-MoSeq (kpmoseq) and Optipose. Optipose is a 3D-only method, hence no results for Optipose are presented for the 2-Mice-2D dataset. **c - e** Example of the imputation of missing coordinates of one keypoint by DISK and other tested methods (Mouse FL2 dataset). **f - g** RMSE with respect to the gap length between tested architectures (**f**) and methods (**g**) on the 2-Fish dataset (same color scheme as panels a and b). **h - i** Output results of DISK and other methods on real gaps of one 1-minute recording from the Mouse FL2 dataset. **i** is a temporal zoom of **h**.

Figure 2: Comparison of different architectures (a & b) and of different methods (c & d) a MPJPE (normalized units), **b** PCK@0.01, **c** MPJPE (original units, millimeters for all datasets except for Human dataset), **d** PCK@0.01. While MPJPE is very close to the RMSE metric, PCK@0.01 displays larger variations between datasets and movement ranges. For example, in DF3D, the fly is maintained at the torso making its movements restrained and of small amplitude.

Figure 3: Test RMSE with respect to the gap length. Panels in the first row compare the different DL architectures and report RMSE in normalized units. Panels in the second row compare the different methods and report RMSE in the original coordinate units (millimeters for all datasets except for the Human dataset). **a** CLB, **b** FL2, **c** DF3D, **d** Human, **e** Rat7M, **f** 2-Mice-2D.

Figure 4: DISK metrics (RMSE, MPJPE, PCK@0.01) for all datasets for each missing keypoint and for all keypoints averaged ("all"): DF3D (keypoints numbered as in the original data (cf github.com/NeLy-EPFL/DeepFly3D), FL2, CLB, Human, Rat7M, 2-Fish, 2-Mice-2D (kp0: nose, kp1: left ear, kp2: right ear, kp3: neck, kp4: left abdomen, kp5: right abdomen, kp6: tail base). (in original units, millimeters except for the Human dataset).

Figure 5: Imputation of switched keypoints.

Figure 5: **a & b** Examples of switched keypoints at test time from the 2-Mice-2D dataset. The blue trajectory represents the ground truth; the dashed grey line is the sample with the switched keypoints given as input to the DISK proba models; the straight grey line is the linear interpolation of the gap. In yellow are shown the trajectory and uncertainty for the DISK proba model trained with the switch data augmentation. In red the DISK model trained with the switch data augmentation. **c** Averaged RMSE, PCK@0.01 and MPJPE for linear interpolation and the two DISK proba models expressed for test samples with ($swap = True$) and without switched keypoints ($swap = False$). Using the switch data augmentation module improves the model’s accuracy in case of switched keypoints, while keeping the accuracy the same for non-switched keypoints. **d & g** Averaged PCK@0.01 for examples as a function of pairs of switched keypoints, **e & h** as a function of gap length and switch length (expressed in frames), and **f & i** as a function of the averaged distance between switched keypoints for the two 2-animals datasets (2-Mice-2D and 2-Fish). All RMSEs and MPJPEs are expressed in mm.

Figure 6: RMSE, MPJPE and PCK@0.01 for the DISK models with and without switch module for **a** CLB, **b** 2-Mice-2D, **c** Rat7M, **d** DF3D, **e** Human, **f** 2-Fish datasets. (original units, millimeters for all datasets except for Human dataset).

Figure 7: Additional plots reporting PCK@0.01 values similar to 5 d-i for the other datasets (original units, millimeters for all datasets except for Human dataset; gap length expressed in frames). As the pairs of keypoints being switched are chosen randomly, blank matrix squares correspond to combinations not drawn in the limited test set.

Figure 8: Analysis of DISK performance with respect to action classes, quantity of movement, periodicity, and gap length, on a - e 2-Mice-2D and f - j Human datasets. a & f Histogram of averaged MPJPE per sample. b & g Averaged MPJPE per sample separated by action class. c & h Averaged MPJPE per sample according to the average overall movement present in the sample. d & i Same according to periodicity. Periodicity is defined as the maximum weight associated with a single frequency after Fourier transform (see Methods section 3.9 "Computation of movement and periodicity per sample"). e & j Same according to gap length (expressed in frames).

Reviewer #1 (Remarks to the Author):

A.-H. See previous review.

I read the comments and questions of the other reviewer. The authors' responses are clear and thoughtful. The additional material that was added to the paper due to the review process is helpful.

We heartily thank the reviewer for helping us improve the manuscript and for the positive feedback.

Some wording & spelling suggestions:

Abstract: "We found that transformer outperforms other architectures..." When using "transformer" as the name of a specific transformer architecture, it should be capitalized: "Transformer." If the authors mean a generic transformer architecture, an article is needed: "We found that a transformer outperforms other architectures..."

As we shorten the abstract under 150 words, we removed the mention to transformer architecture in the abstract.

OptiPose should be consistently spelled with a capital P. The current document uses both OptiPose and Optipose.

We changed all occurrences of OptiPose to the right spelling.

"At test time, to compare the performance of on exactly the same sequences," -> At test time, to compare the performance of the models on exactly the same sequences,

We changed the sentence as suggested by the reviewer.

Remarks on code availability:

As mentioned in the initial review, the provided notebook is easy to follow and covers all relevant use cases. Training a model and running it in inference mode seem straightforward. The code is reasonably well-documented and easy to follow. The authors have also provided an FAQ that covers what the hyperparameters do and how to change them.

We thank the reviewer for their positive feedback on the code bases.

Reviewer #2 (Remarks to the Author):

This edited manuscript addressed all of the questions I raised during previous reviews. I appreciate the thorough rebuttal and effort in implementing the MBI comparison and in all of the additional analyses that addressed my questions about generated datasets, keypoint switching, and in general clarification on the implementation of DISK and how it differs

and is a complimentary tool to pose tracking. I also appreciate the well-commented code repositories.

We heartly thank the reviewer for helping us improve the manuscript and for his/her positive feedback.